# Adaptive Decision-Making for Optimization of Safety-Critical Systems: The ARTEO Algorithm

## Abstract

Real-time decision-making in uncertain environments with safety constraints is a common problem in many business and industrial applications. In these problems, it is often the case that a general structure of the problem and some of the underlying relationships among the decision variables are known and other relationships are unknown but measurable subject to a certain level of noise. In this work, we develop the ARTEO algorithm by formulating such real-time decision-making problems as constrained mathematical programming problems, where we combine known structures involved in the objective function and constraint formulations with learned Gaussian process (GP) regression models. We then utilize the uncertainty estimates of the GPs to (i) enforce the resulting safety constraints within a confidence interval and (ii) make the cumulative uncertainty expressed in the decision variable space a part of the objective function to drive exploration for further learning – subject to the safety constraints. We demonstrate the safety and efficiency of our approach with two case studies: optimization of electric motor current and real-time bidding problems. We further evaluate the performance of ARTEO compared to other methods that rely entirely on GP-based safe exploration and optimization. The results indicate that ARTEO benefits from the incorporation of prior knowledge to the optimization problems and leads to lower cumulative regret while ensuring the satisfaction of the safety constraints.

## 1 INTRODUCTION

Sequential decision-making under uncertainty frequently involves unknown yet measurable noisy functions, which are sequentially evaluated to determine the optimal decisions. In this stochastic optimization context, decisions are initially heuristic-driven, with each resulting reward used to inform subsequent decisions. Over time, this reduces decision uncertainty. While exploration can optimize decisions by offering more information, it can also be costly in various applications. The challenge is to balance the exploration of new decision points with the exploitation of prior experiences, a subject well-explored in literature, especially within multi-armed bandit (MAB) approaches using confidence bounds (Bubeck et al., 2012).

Bayesian optimization (BO) has emerged as a robust technique for the optimization of unknown or expensive-to-evaluate functions, making use of probabilistic models such as Gaussian processes (GPs) to predict function values and guide the search process (Brochu et al., 2010; Shahriari et al., 2015). Within the broader context of BO, the introduction of confidence bounds, particularly the upper confidence bound (UCB), offered a mechanism to balance the exploration-exploitation trade-off (Lai & Robbins, 1985). This insight led to the development of several UCB algorithms specifically designed for stochastic bandits (Lattimore & Szepesvári, 2020). Building on these foundations, efficient algorithms have been constructed for bandit problems under certain regularity conditions (Dani et al., 2008; Bubeck et al., 2009). Later work by Srinivas et al. (2010) divided the stochastic optimization challenge into two main objectives: the estimation of unknown functions and their subsequent optimization. Central to their approach was the employment of kernel methods and GPs for the modelling of the reward function (Rasmussen, 2004). While Bayesian optimization (BO) has proven effective in optimizing expensive or unknown functions, traditional BO often overlooks safety constraints, thus highlighting the necessity for safe BO algorithms.

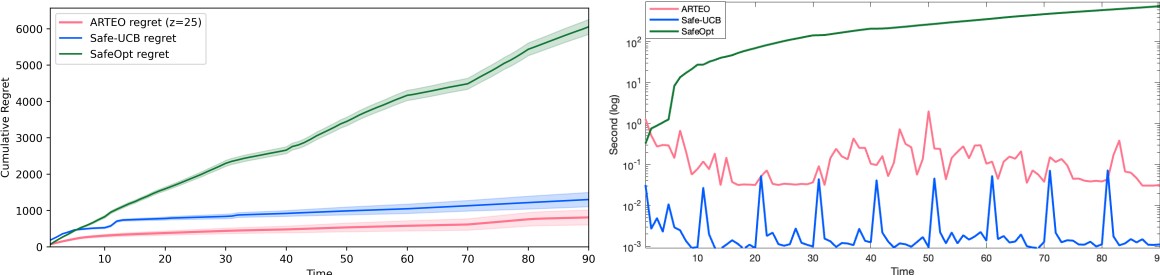

Figure 1: **(left)** Cumulative regret of ARTEO and Safe-UCB with 50 different safe seeds. Shaded area represents $\pm 0.1$ standard deviation. **(right)** Comparison of time spent to complete first 90 timesteps of the reference signal in Figure 2.

In safety-critical systems, direct exploration can be hazardous. These systems, like chemical process plants, are safety-critical due to potential human or environmental risks. Therefore, optimization in such contexts requires algorithms that permit only safe exploration by adhering to safety constraints. This work characterizes feasible decision points as those meeting the safety constraints of the specific problem.

Safe exploration of black-box functions has been analyzed in-depth in safe BO literature (Sui et al., 2015; Schreiter et al., 2015; Turchetta et al., 2016; Wachi et al., 2018; Sui et al., 2018; Turchetta et al., 2019). Sui et al. (2015) introduced the SafeOpt algorithm, which allows for the safe optimization of unknown functions. This and similar algorithms have been effectively applied to numerous control and reinforcement learning challenges (Berkenkamp et al., 2016; Kabzan et al., 2019). Recent works addressed the limitations of safe BO such as scalability issues for high-dimensional decision sets (Kirschner et al., 2019) or reaching a local optimum when the reachable decision space from the initial condition is disjoint from the decision space including the global optimum Sukhija et al. (2023).

Most safe BO algorithms consist of two elements: expanding current safe decision space (expanders) and optimizing within that space (maximizers) (Sui et al., 2018; Berkenkamp et al., 2016; Kabzan et al., 2019; Sukhija et al., 2023). Continuous exploration for creating an expanders set is not viable in contexts like industrial processes as any deviation from optimization goals can result in significant losses and optimization goals of the system should be pursued even during exploration. For example, any deviation from target satisfaction in an industrial process may lead to unsalable products and hence to economic losses. This highlights the need for adaptive exploration tailored to context-dependent needs.

In numerous real-world situations, optimization problems possess partial information derived from fundamental principles or domain insights. This suggests the possibility of moving beyond a pure black-box model and leveraging the available first-principle-driven knowledge to enhance optimization efforts. While the concept of grey-box BO, which combines white-box models and unknown (black-box functions) ones, has been covered to some extent (Astudillo & Frazier, 2019; 2021; 2022; Lu & Paulson, 2023), the main challenge lies in the safe exploration of these grey-box functions. Notably, ensuring safety within this grey-box framework is a research area that remains less explored compared to conventional black-box BO.

In this work, we formulate a safe optimization problem where the objective and safety constraint functions include first-principle-driven functions with known explicit forms besides black-box functions. Then, we propose a novel framework for Adaptive Real-Time Exploration and Optimization (ARTEO) of the formulated problem in 2. Our contributions are outlined below:

- We formulate a safe optimization problem where the objective and safety constraint functions are composed of known functions with known explicit forms and black-box functions. Under certain assumptions, we extend the safety guarantees of BO for black-box functions to a grey-box optimization, which is prevalent in industrial settings.

- Departing from having separate iterations for the expanders-maximizers in safe BO, we propose an approach that considers both simultaneously. By incorporating uncertainty directly into the objective

functions, we can control the balance between exploration and exploitation through a hyperparameter that allows for adaptive exploration. Notably, setting this hyperparameter to zero halts exploration, ensuring safety in cases where prior explorative decisions resulted in constraint violations. This approach eliminates the need for a separate acquisition function to guide the exploration.

- We empirically validate our approach on both low-dimensional (2-decision variables) and high-dimensional (200-decision variables) safety-critical case studies. Our results indicate successful optimization without safety violations. Furthermore, by empirically comparing cumulative regret against established methods such as Safe-UCB and SafeOpt, we demonstrate that our approach yields lower cumulative regret, and emphasize its efficacy as seen in Figure 1.

## 2 PROBLEM STATEMENT AND BACKGROUND

We consider the safe optimization of grey-box functions in a time-varying setting. We have a cost function that we seek to minimize without violating the safety constraints. This objective function is composed of functions known with explicit forms and black-box functions of the decision variables for a task in iteration $t = \{1, \ldots, T\}$ in an environment with the time-dependent behaviour of $C_t$ remaining unspecified (unknown) across iterations. At each iteration, we choose the decision point $x^*$ that minimizes the economic cost of our decisions while helping the more accurate modelling of black-box functions through acquiring information at the most uncertain points where $x \in D$ is a real-valued vector of decision points and $D \subset \mathbb{R}^n$ is a known compact subset of $\mathbb{R}^n$. Next, we define the necessary concepts and formulate our problem in interest.

Consider a set of functions with known explicit forms (white-box models), denoted as $\Delta_j : \mathbb{R}^n \to \mathbb{R}$ for $j \in \mathcal{J}$, where $\mathcal{J}$ is the index set for all functions with known explicit forms. Conversely, the black-box functions $p_i : \mathbb{R}^n \to \mathbb{R}$ for $i \in \mathcal{I}$ are unknown, and we obtain noisy observations for their evaluations at the decision point $x$, $y_i = p_i(x) + \epsilon_i$, within the environment where our experiments take place. Here, $\epsilon_i$ is assumed to be independent and identically distributed (i.i.d) $R$-sub-Gaussian noise with a fixed constant $R \geq 0$ (Agrawal & Goyal, 2013). Given a grey-box function $C : \mathbb{R}^m \to \mathbb{R}$ that is a composition of white-box and black-box functions, and safety constraints modelled through grey-box functions $g_a : \mathbb{R}^m \to \mathbb{R}$ for $a \in \{1, \ldots, A\}$, where $A$ is the number of constraints, our safe optimization problem at iteration $t$ can be formulated as:

$$\begin{aligned} x^* =& \arg\min_{x \in D} C_t(v(x)) \\ \text{subject to } & g_{a,t}(v(x)) \leq h_{a,t}, \quad \forall a \in \{1, \ldots, A\} \end{aligned} \tag{1}$$

where $h_{a,t}$ denotes the safety threshold for the safety constraint $g_{a,t}$ and $v(x) : \mathbb{R}^n \to \mathbb{R}^m$ constructs a vector from the evaluations of the functions $\Delta_j$ and $p_i$ at $x$, specifically $v(x) = (\Delta_1(x), \ldots, \Delta_{|\mathcal{J}|}(x), p_1(x), \ldots, p_{|\mathcal{I}|}(x))$ such that $|\mathcal{J}| + |\mathcal{I}| = m$.

### 2.1 Gaussian processes

Gaussian processes are non-parametric models which are useful for regression and not data-intensive compared to other common regression techniques in machine learning. GPs are fully specified by a mean function $\mu(x)$ and a kernel $k(x, x')$ which is a covariance function and specifies the prior and posterior in GP regression (Rasmussen, 2004). The covariance function of GP's makes them commonly used for the estimation of black-box functions in safe learning literature The goal is to learn the black-box functions $p_i, i \in \mathcal{I}$ over the decision set $D$ by using GPs to solve the optimization problem in (1). Assuming having a zero mean prior, the posterior over $p$ follows $\mathcal{N}(\mu_T(x), \sigma_T^2(x))$ with,

$$\begin{aligned} \mu_T(x) =& k_T(x)^T (K_T + \sigma^2 I)^{-1} y_T \\ k(x, x') =& k(x, x') - k_T(x)^T (K_T + \sigma^2 I)^{-1} k_T(x') \\ \sigma^2(x) =& k_T(x, x) \end{aligned} \tag{2}$$

where $k_T(x) = [k(x_1, x), \ldots, k(x_T, x)]$, and $K_T$ is the positive definite kernel matrix $[k(x, x')]_{x,x' \in \{x_1, \ldots, x_T\}}$. By using GPs, we can define estimated $\hat{p}_i$ at $x$ with a mean $\mu_{\hat{p}_i}(x)$ and standard deviation $\sigma_{\hat{p}_i}(x)$.

## 2.2 Regularity assumptions

In this study, we proceed without specific knowledge of the time-dependent changes of $C_t$ and $g_{a,t}$ across iterations $t$. However, we need to make some assumptions that are applied uniformly across all iterations to provide safety with high probability at decision points (Srinivas et al., 2010; Sui et al., 2015; Berkenkamp et al., 2016; Sui et al., 2018). For simplicity, we will proceed with a single safety constraint ($A = 1$) and one unknown function. The generalizability of the proposed algorithm to multiple black-box functions and safety constraints is discussed in Appendix B. The discussions in this section are applicable to any cost function $C_t$, safety function $g_{a,t}$, and black-box function $p_i$, across all iterations $t = \{1, \ldots, T\}$. We represent them as $C$, $g$ and $p$ without index. Similar to (Schreiter et al., 2015; Sui et al., 2015; 2018; Berkenkamp et al., 2016; Kabzan et al., 2019), we make some regularity assumptions common in safe optimization literature.

**Assumption 2.1.** The decision set $D$ is compact (Hanche-Olsen & Holden, 2010).

**Assumption 2.2.** The cost function $C$ and safety function $g$ have explicitly known functional forms and they are nested functions of function known with explicit form $\Delta$, which is assumed to be continuous over $D$, besides black-box function $p$.

**Assumption 2.3.** Noisy observations are obtained for each black-box function $p_i : \mathbb{R}^n \to \mathbb{R}$, $i \in \mathcal{I}$, represented as $y_i = p_i(x) + \epsilon$, where $\epsilon$ is characterized as independent and identically distributed (i.i.d.) $R$-sub-Gaussian noise with a fixed constant $R \geq 0$ (Agrawal & Goyal, 2013).

**Assumption 2.4.** $p_i \in \mathcal{H}_{k_i}, i \in \mathcal{I}$ where $k_i : \mathbb{R}^{n_i} \times \mathbb{R}^{n_i} \to \mathbb{R}$, $i \in \mathcal{I}$ are positive-semidefinite kernel functions (Rasmussen, 2004), $\mathcal{H}_{k_i}$ are the corresponding reproducing kernel Hilbert spaces (RKHS), and $n_i$ is the dimension of the input variables for the function $p_i$.

The RKHS is formed by Lipschitz-continuous functions and the inner product of functions in RKHS follows the reproducing property: $\langle p_i, k_i(x, \cdot) \rangle_{k_i} = p_i(x)$ for all $p_i \in \mathcal{H}_{k_i}, i \in \mathcal{I}$. The smoothness of a function in RKHS with respect to kernel function $k_i, i \in \mathcal{I}$ is measured by its RKHS norm $\|p_i\|_{k_i} = \sqrt{\langle p_i, p_i \rangle_{k_i}}$ and for all functions in $\mathcal{H}_{k_i}$, $\|p_i\|_{k_i} < \infty$ (Scholkopf & Smola, 2001).

**Assumption 2.5.** There exits a known bound $B_i$ for the RKHS norm of the unknown function $p_i : \|p_i\|_{k_i} < B_i, i \in \mathcal{I}$.

We use this bound $B_i$ to control the confidence interval (CI) width later in Equation (4). In most cases, we are not able to compute the exact RKHS norm of the unknown function $p_i$ as stated by previous studies (Jiao et al., 2022). Alternative approaches are choosing a very large $B_i, i \in \mathcal{I}$, or obtaining an estimate for $B_i, i \in \mathcal{I}$ by guess-and-doubling. It is possible to apply hyperparameter optimization methods to optimize $B_i, i \in \mathcal{I}$ where data is available offline (Berkenkamp et al., 2019). The choice of $\beta_t$ in ARTEO is explained later in Section 4. Next, we introduce a lemma for the continuity of $g$.

**Lemma 2.6.** *[adapted from Hewitt (1948)] Let $g : \mathbb{R}^m \to \mathbb{R}$ be a grey-box function composed of function known with explicit form $\Delta : \mathbb{R}^n \to \mathbb{R}$ and black-box function $p : \mathbb{R}^n \to \mathbb{R}$. $p$ and $\Delta$ are continuously defined in domain $D$. Given $p$ and $\Delta$ are continuous in $D$, any $g$ function that is formed by an algebraic operation over two functions $p$ and $\Delta$ is also continuous in $D$.*

Following 2.6, the continuity assumption holds for the safety function $g$ since it is formed by continuous functions $\Delta$ and $p$ under the Assumption 2.2. Next, we define the relationship between $g$ and $p$.

**Definition 2.7.** [adapted from Kimeldorf & Sampson (1978)] *(monotonically related)* A function $\phi(\cdot, \pi(x))$ is monotonically related to $\pi(x)$ if $\phi$ and $\pi$ are continuous in $D$ and for any $x, y \in D$ such that $\pi(x) \leq \pi(y) \Rightarrow \phi(\cdot, \pi(x)) \leq \phi(\cdot, \pi(y))$.

**Definition 2.8.** [adapted from Kimeldorf & Sampson (1978)] *(inversely monotonically related)* A function $\phi(\cdot, \pi(x))$ is inversely monotonically related to $\pi(x)$ if $\phi$ and $\pi$ are continuous in $D$ and for any $x, y \in D$ such that $\pi(x) \leq \pi(y) \Rightarrow \phi(\cdot, \pi(x)) \geq \phi(\cdot, \pi(y))$.

**Assumption 2.9.** $g$ is monotonically related or inversely monotonically related to $p$ as in Definition 2.7 and Definition 2.8.

Assumption 2.9 allows us to reflect confidence bounds of $\hat{p}$ to $g$. The continuity assumptions and ability to provide confidence bounds for $\hat{p}$ depend on which model is used to estimate $p$. In this paper, we choose GPs

which are related to RKHS through their positive semidefinite kernel functions (Sriperumbudur et al., 2011) that allow us to construct confidence bounds in a safe manner later.

## 2.3 Confidence bounds

In ARTEO, the solver uses the CI which is constructed by using the standard deviation of conditioned GP on previous observations to decide the feasibility of a chosen point $x$. Hence, the correct classification of decision points in $D$ relies on the confidence-bound choice. Under the regularity assumptions stated in Section 2.2, Theorem 3 of Srinivas et al. (2010) and Theorem 2 of Chowdhury & Gopalan (2017) proved that it is possible to construct confidence bounds which include the true function with probability at least $1 - \delta$ where $\delta \in (0, 1)$ on a kernelized multi-armed bandit problem setting with no constraints. A recent study proposed less conservative yet efficient tighter confidence bounds for GPs (Fiedler et al., 2021). Moreover, as shown by Theorem 1 in (Sui et al., 2018), this theorem is applicable to multi-armed bandit problems with safety constraints. Hence, we can state that the probability of the true value of $p$ at the decision point $x$ is included inside the confidence bounds in iteration $t$:

$$P\big[|p(x) - \mu_{t-1}(x)| \leq \beta_t \sigma_{t-1}(x)\big] \geq 1 - \delta, \ \forall t \geq 1 \tag{3}$$

where $\mu_{t-1}(x)$ and $\sigma_{t-1}(x)$ denote the mean and the standard deviation at $x$ from a GP at time $t$, which is conditioned on past $t - 1$ observations to obtain the posterior. $\delta$ is a parameter that represents the failure probability in Equation (3). $\beta_t$ controls the width of the CI and satisfies Equation (3) when:

$$\beta_t = B + R\sqrt{2(\gamma_{t-1} + 1 + \ln(1/\delta))} \tag{4}$$

where the noise in observations is $R$-sub-Gaussian and $\gamma_{t-1}$ represents the maximum mutual information after $t - 1$ iterations. $\gamma_t$ is formulated as:

$$\gamma_t = \max_{|x_t| \leq t} I(\hat{p}; y_t) \tag{5}$$

where $y_t$ represents the noisy observations of $\hat{p}(x)$ at the decision point at time $t$. $I(\cdot)$ denotes the mutual information as such:

$$I(p; y_t) = 0.5 \log |\mathrm{I} + \sigma^{-2} K_t| \tag{6}$$

Equation (3) gives us the confidence bounds of $\hat{p}$. However, in order to establish safety with a certain probability, we need to obtain confidence bounds of $g(x)$ for each iteration. To expand Equation (3) for $g$, we provide the following lemma.

**Lemma 2.10.** *Let $\hat{p}^L = \mu_t(x) - \beta_t \sigma_t(x)$ and $\hat{p}^U = \mu_t(x) + \beta_t \sigma_t(x)$ where $\hat{p}^L \leq \hat{p}(x) \leq \hat{p}^U$. Given $g$ is monotonically related to $\hat{p}$, $g$ is a real-valued function $g(\Delta(x), \hat{p}(x))$ with a known explicit functional form of $\Delta(x)$, and $\hat{p}^L \leq \hat{p}(x) \leq \hat{p}^U$:*

- *if $g$ is monotonically related to $\hat{p} \Rightarrow g(\Delta(x), \hat{p}^L) \leq g(\Delta(x), \hat{p}(x)) \leq g(\Delta(x), \hat{p}^U)$.*

- *if $g$ is inversely monotonically related to $\hat{p} \Rightarrow g(\Delta(x), \hat{p}^U) \leq g(\Delta(x), \hat{p}(x)) \leq g(\Delta(x), \hat{p}^L)$.*

Now, we present the main theorem that establishes satisfaction of constraints in problem (1) with high probability based on regularity assumptions and Lemma 2.10. The proofs for Lemma 2.10 and Theorem 2.11 are given in Appendix A.

**Theorem 2.11.** *Let Assumptions 2.1, 2.2, 2.3, 2.4, 2.5 and 2.9 hold. The maximum and minimum values of $g(\Delta(x), \hat{p}(x))$ lie on the upper and lower confidence bounds of the Gaussian process obtained for $\hat{p}(x)$ which are computed as in Lemma 2.10. For a chosen $\beta_t$ and allowed failure probability $\delta$ as in Equation (4),*

- *if $g$ is monotonically related to $\hat{p} \Rightarrow P\big[g_t(\Delta(x), \hat{p}^L) \leq g_t(\Delta(x), \hat{p}(x)) \leq g_t(\Delta(x), \hat{p}^U)\big] \geq 1 - \delta, \forall t \geq 1$*

- *if $g$ is inversely monotonically related to $\hat{p} \Rightarrow P\big[g_t(\Delta(x), \hat{p}^U) \leq g_t(\Delta(x), \hat{p}(x)) \leq g_t(\Delta(x), \hat{p}^L)\big] \geq 1 - \delta, \forall t \geq 1$*

# 3 ARTEO ALGORITHM

We develop the ARTEO algorithm for safety-critical environments with high exploration costs. At each iteration, the algorithm updates the posterior distributions of GPs with previous noisy observations as in Equation (2) and provides an optimized solution for the desired outcome based on how GPs model the black-box functions. It does not require a separate training phase, instead, it learns during normal operation. The details of safe learning and optimization are given next.

## 3.1 Safe learning

The decision set $D$ is defined for decision variables as satisfying Assumption 2.1. For each black-box function $p_i, i \in \mathcal{I}$, a GP prior and initial "safe seed" set is introduced to the algorithm. The safe seed set $S_0$ includes at least one safe decision point with the true value of the safety function at that point satisfying the safety constraint(s). As in many published safe learning algorithms (Sui et al., 2015; 2018; Turchetta et al., 2019), without a safe seed set, an accurate assessment of the safety of any points is difficult. Each iteration of the algorithm could be triggered by time or an event. After receiving the trigger, the algorithm utilizes the past noisy observations $y_i(x)$ in a form $(x, y_i(x))$ to obtain the GP posterior for $p_i, i \in \mathcal{I}$ and use in the optimization of the objective function, which includes the cost of decision and uncertainty. For the first iteration, safe seed sets are given as past observations.

Now, we formulate the objective function $f_t$ in ARTEO which includes the cost of decisions $C_t$ and uncertainty $U_t$ which is quantified at the evaluated point $x$ as:

$$U_t(x) = \sum_{i \in \mathcal{I}} \sigma_{\hat{p}_i}(x) \tag{7}$$

where $\sigma_{\hat{p}_i}(x)$ is the standard deviation of $\mathcal{GP}_i$ at $x$ for the unknown function $p_i$ in the iteration $t$ and is obtained in Equation (2). It is incorporated into the $f_t$ by multiplying by an adjustable parameter $z$ as next:

$$f_t(v(x)) = C_t(v(x)) - zU_t(x) \tag{8}$$

where $C_t(v(x))$ represents the cost of the decision based on the evaluations of $\Delta_j, j \in \mathcal{J}$, and $p_i, i \in \mathcal{I}$, with $v(x) = (\Delta_1(x), \ldots, \Delta_{|\mathcal{J}|}(x), p_1(x), \ldots, p_{|\mathcal{I}|}(x))$ at the chosen decision point. In our experiments, the priority of the algorithm is optimizing the cost of decisions under given constraints. Hence, the uncertainty weight $z$ remains zero until the environment becomes available for exploration. Until that time, the algorithm follows optimization goals and learns through changes in the optimization goals such as operating in a different decision point to satisfy a changed demand of motor current in our first case study.

The exploration is controlled by the $z$ hyperparameter. Choosing the optimal value of the hyperparameter is challenging in the online learning setup where complete information is not available to do simulation and several approaches have been proposed recently to overcome this issue (Letham & Bakshy, 2019; De Ath et al., 2021). We initialize $z = 0$ and then we train $\mathcal{GP}_{hyp}$ to capture the uncertainty in the optimized decisions over iterations and the relationship of it with set $z$ values at previous iterations. We update $\mathcal{GP}_{hyp}$ with the latest information at the end of each iteration and obtain a new $z$ value which is chosen by the lowest confidence bound of $\mathcal{GP}_{hyp}$ predictions in the interval of $z_{lb}$ and $z_{ub}$. If a constraint is violated due to the inaccuracy of predictions $\mathcal{GP}_i$ at high uncertainty points, we set $z = 0$ to stop exploration and make the optimization stable in the next iteration. The pseudocode of the algorithm is given in Algorithm 1.

We investigate the impact of different approaches for selecting the hyperparameter $z$ during exploration. Specifically, we analyze the effects of using a constant $z$ value, choosing $z$ based on uncertainty using $\mathcal{GP}_{hyp}$, and using an instantaneous regret-based $\mathcal{GP}_{hyp}$ approach. These analyses are presented in Section 4. Additionally, we share the results of offline hyperparameter optimization methods for situations where simulation data is available in Appendix C.1.3.

## 3.2 Optimization

The RTO incorporates the posterior of the Gaussian process into decision-making by modelling the black-box functions by using the mean and standard deviation of GPs. The cost function is the objective function

---

**Algorithm 1** ARTEO

---

1: **Input:** Decision set $D$, GP priors for each $\mathcal{GP}_i$, safe seed set for each $\mathcal{GP}_i$ as $S_{i,0}$, cost function $f$, safety function $g$, safety threshold $h$, lower $z_{lb}$ and upper bound $z_{ub}$ for hyperparameter search
2: $z \leftarrow 0$ $\qquad\qquad\qquad$ # Initialize $z$
3: $r_0 \leftarrow \sum_{i=1}^m S_{i,0}$ $\qquad\qquad$ # Initialize regret
4: $H_0 \leftarrow \{(z, r_0, t = 0)\}$ $\qquad$ # Initialize the $z$ and $r$ data to train $\mathcal{GP}_{hyp}$
5: Train $\mathcal{GP}_{hyp}$ on $H_0$ $\qquad\quad$ # Initialize $\mathcal{GP}_{hyp}$
6: **for** $t = 1, ..., T$ **do**
7: $\quad$ **for** $i = 1, ..., |\mathcal{I}|$ **do**
8: $\quad\quad$ Update $\hat{p}_i$ by conditioning $\mathcal{GP}_i$ on $S_{i,t-1}$
9: $\quad$ **end for**
10: $\quad x^* \leftarrow \arg\min_{x_i \in D_i} f_t$ s.t. $g_t$
11: $\quad$ **for** $i = 1, ..., |\mathcal{I}|$ **do**
12: $\quad\quad$ Obtain noisy observation $y_{i,t}$ for chosen $x^*$
13: $\quad\quad S_{i,t} \leftarrow S_{i,t-1} \cup \{(x^*, y_{i,t})\}$
14: $\quad$ **end for**
15: $\quad$ Calculate a task-dependent regret for $r_t$ using chosen $x^*$ for $f_t$ s.t. $g_t$
16: $\quad H_t \leftarrow H_{t-1} \cup \{(z, r_t, t-1)\}$
17: $\quad$ Update $\mathcal{GP}_{hyp}$ with $H_t$
18: $\quad$ **if** t > 1 and $x^*$ does not violate any constraints **then**
19: $\quad\quad$ Set $z$ to a value that has the lowest confidence bound in $\mathcal{GP}_{hyp}$ predictions over the interval $[z_{lb}, z_{ub}]$
20: $\quad$ **else**
21: $\quad\quad z \leftarrow 0$
22: $\quad$ **end if**
23: **end for**

---

in the RTO formulation and the safety thresholds are constraints. In the cost function, the mean of GP posterior of each unknown function is used to evaluate the cost of decision and the standard deviation of GPs is used to measure uncertainty as in Equation (7). In the safety function, the standard deviation of the GP posterior of each unknown function is used to construct confidence bounds and then these bounds are used to assess the feasibility of evaluated points. The optimizer solves the minimization problem under safety constraints within the defined decision set of each decision variable. Any optimization algorithm that could solve the given problem can be used in this phase.

### 3.3 Related work

Our problem setting intersects with several research domains such as real-time optimization (RTO), safe BO, and time-varied BO (TVBO). We further elaborate on the distinct features of our approach here.

**RTO.** If we know the closed form of $p_i(x)$, we can solve the problem as an optimization problem. For instance, we could use the concept of RTO from the process control domain (Naysmith & Douglas, 2008) and use the approach proposed by Petsagkourakis et al. (2021). However, the explicit form of black-box function $p_i(x)$ is unknown for all black-box functions where $i \in \mathcal{I}$. Thus, at every iteration $t$, we first solve an estimation problem to find $p$, then solve the optimization problem (1). Solving an optimization problem by combining estimation then optimization is a common approach (Zhang et al., 2022; Fu & Levine, 2021). However, few approaches quantify the uncertainty inherent in the estimation of $p_i$, $i \in \mathcal{I}$. In this work, we propose to estimate $p_i, i \in \mathcal{I}$ using Gaussian processes and use regularity assumptions of Gaussian processes to ensure safety.

**Safe BO.** Safe BO is a critical domain for optimization under uncertainty, particularly when it is vital to adhere to safety constraints. The introduction of the SafeOpt algorithm by Sui et al. (2015) marked a pivotal point in the Safe BO domain. This algorithm and its variations have been effectively utilized in various control and reinforcement learning challenges, demonstrating the practical applications of safe BO (Berkenkamp et al., 2016; Kabzan et al., 2019).

Further developments in safe BO include addressing scalability issues in high-dimensional decision sets (Kirschner et al., 2019) and overcoming the challenge of reaching local optima when the reachable decision space from the initial condition is disjoint from the global optimum space (Sukhija et al., 2023). Additionally, significant contributions from Schreiter et al. (2015), Turchetta et al. (2016), Wachi et al. (2018), Sui et al. (2018), and Turchetta et al. (2019) have enriched the understanding, studied the limitations and applications of safe BO, focusing on the safe exploration of black-box functions.

One common solution in Safe BO for safe optimization under uncertainty is that modelling objective function $f$ and constraint function $g$ as black-box functions and sampling information from decision points that maximize the objective function value (maximizers) or safely exploring highly uncertain points with expanding the set of explored decision points (expanders) (Sui et al., 2015). ARTEO does not need to choose to solve the optimization problem for either maximizers or expanders since it considers the exploration and exploitation simultaneously in objective function through balancing the cost of decisions and uncertainty in the environment with a hyperparameter.ARTEO does not need to choose to solve the optimization problem for either maximizers or expanders since it considers the exploration and exploitation simultaneously in objective function through balancing the cost of decisions and uncertainty in the environment with a hyperparameter. Furthermore, (Astudillo & Frazier, 2019) shows how using grey-box formulation by leveraging the available information regarding function known with explicit form structure outperforms pure black-box-based BO methods.

**TVBO**: In the context of our study, the optimal decision inherently varies with time, $t$. This variation is attributed to the time-sensitive cost of decisions. For instance, our framework might be interpreted as a control problem where any shift in the setpoints to be followed leads to an increase in $t$. Consequently, every time there is such a change, the problem (1) needs to be addressed anew, accounting for the new setpoints at $t + 1$. The event-driven nature of our objective function bears a resemblance to the scenario addressed by TVBO. While many TVBO techniques operate under the assumption that the rate of change in the objective function is both known and constant (Bogunovic et al., 2016; Su et al., 2018; Parker-Holder et al., 2020; 2021), a recent contribution by Brunzema et al. (2022) proposed a method that does not rely on a fixed rate of change. In their approach, every new event trigger prompts a reset of the sampled data set. Conversely, our method offers two significant distinctions from existing TVBO algorithms: we do not mandate prior knowledge of the rate of change, offering more flexibility, and unlike the reset approach, we retain past observation data as we transition to subsequent iterations. In this work, we operate on the belief that past data retains relevance and utility in learning the black-box function. The decision regarding which data to retain, discard, or remember weakly depends on the characteristics of the experiments under study. For example, in some real-world problems, it is necessary to consider data shifts that may render past data obsolete. We leave the detailed investigation of such specialized cases for future work.

### 3.4 Complexity

In each iteration, ARTEO updates GP models by conditioning GPs on past observations and finds a feasible solution. The overall time complexity of each run of the algorithm is the number of iterations $t$ times the time complexity of each iteration. The first computationally demanding step in the ARTEO is fitting GPs on safe sets. The time complexity of training a full GP, i.e. exact inference, is $\mathcal{O}((t-1)^3)$ due to the matrix inversion where $t - 1$ is the number of past observations at iteration $t$. (Hensman et al., 2013). It is possible to reduce it further by using low-rank approximations which is not in the scope of our work (Chen et al., 2013; Liu et al., 2020). We introduce an individual GP for each unknown function $p_i, i \in \mathcal{I}$, so the total complexity of GP calculations is $\mathcal{O}(|\mathcal{I}|(t-1)^3)$ where $|\mathcal{I}|$ represents the number of black-box functions.

The next demanding step is nonlinear optimization. The computational demand of RTO depends on the chosen optimization algorithm and the required $s$ number of steps to converge. The most computationally expensive step is the LDL factorization of a matrix with a $\mathcal{O}((t-1)^3)$ complexity in both used optimization algorithms in this paper (Schittkowski, 1986; Potra & Wright, 2000). Hence, the complexity of RTO becomes $\mathcal{O}(s(t-1)^3)$. In our implementation, $s \gg |\mathcal{I}|$, so, the time complexity of each iteration in ARTEO scales with the RTO complexity. Therefore, the overall time complexity of one iteration of ARTEO is $\mathcal{O}(s(t-1)^3) \approx \mathcal{O}(st^3)$. The memory complexity of the algorithm is $\mathcal{O}(|\mathcal{I}|(t-1)^2) \approx \mathcal{O}(|\mathcal{I}|t^2)$, which is dominated by matrix storage in GPs and optimization.

### 3.5 Limitations

ARTEO's theoretical safety guarantee is based on the assumption of monotonicity between the safety constraint and black-box functions. However, it's worth noting that monotonic relationships in functions are not uncommon in practical settings. For instance, Arellano-Garcia & Wozny (2009) demonstrates applications from the process industry which have safety constraint functions that are strictly monotonic with respect to some variables. Furthermore, Gupta et al. (2016) investigates methods to learn models where the data exhibits monotonic relationships. Lastly, Nieves Avendano et al. (2022) models a case study where the constraints are monotonic and surveys other works concerned with monotonic constraints and functions. These references, along with our own case studies, serve to demonstrate that such monotonic relationships are not just theoretical constructs but are encountered in actual, practical scenarios. While these are just a handful of examples, they underscore the existence of this relationship in diverse settings.

Furthermore, Assumption 2.5 is a strong assumption. However, this assumption is very common in safe BO literature (Sui et al., 2015; 2018; Sukhija et al., 2023) and the successful applications in (Berkenkamp et al., 2016; Jiao et al., 2022) and our experiments demonstrate that it is possible to achieve optimized decisions with minimum safety violations even when this assumption is relaxed and the confidence bounds of GPs have been directly used when computing CI for estimations.

The implementation of ARTEO shares a common limitation amongst algorithms using constraint-based solvers: the initialization problem for starting the optimization (Ivorra et al., 2015). To address this, we use safe seeds as a feasible starting point for the optimization routine and leverage $x^*$ in $t-1$ as an initial guess for the subsequent iteration $t$. However, if $x^*$ in $t-1$ violates the safety constraints (may happen with a probability of $\delta$), the success of the optimization at time $t$ depends on whether the chosen solver can take an infeasible guess as a starting point. Even though many solvers can handle infeasible starting points, they are mostly local solvers which means that if we start far from the actual solution we may reach only a local minimum and it may take a significant amount of time to converge to a solution. Additionally, the time complexity of ARTEO may cause it to be unsuitable for environments that require faster results.

## 4 EXPERIMENTS

In this section, we evaluate our approach on two applications: an electric motor current optimization and online bid optimization. The former problem is introduced by Traue et al. (2022) and the latter one by Liao et al. (2014). We developed the first case study with Matlab 2022a and the second one with Python on an M1 Pro chip with 16 GB memory. The GitHub link is available in the supplementary material.

### 4.1 Electric motor current optimization

In this case study, we implement ARTEO to learn the relationship between torque and current in Permanently-Excited Direct Current Motors (PEDCMs), which have a positively correlated torque-current relationship. We develop the simulation with two PEDCMs as explained in Appendix C.1.1 in Gym Electric Motor (GEM) (Traue et al., 2022). Then, the environment is simulated, and sample data points of torque and current are collected as ground truth in our algorithm.

#### 4.1.1 ARTEO implementation to electric motor current optimization

We put the electric motor current optimization into our framework by following a reference current signal for alternator mode operation where the functions that describe the current at a given torque as a black-box function and unknown to ARTEO. In this case study, ARTEO can only control the torque of the machines to follow the reference current signal. Hence, the algorithm needs to learn the relationship between the applied torque and obtained current for each machine and then determine the torque for each motor to minimize the difference between the total expected current and a given total current reference. The objective function equation 8 is defined as

$$f_t(x) = \left[ Cr_t - \sum_{i=1}^{2} \mu_{TC_i,t}(x) \right]^2 - z \sum_{i=1}^{2} \sigma_{TC_i,t}(x) \tag{9}$$

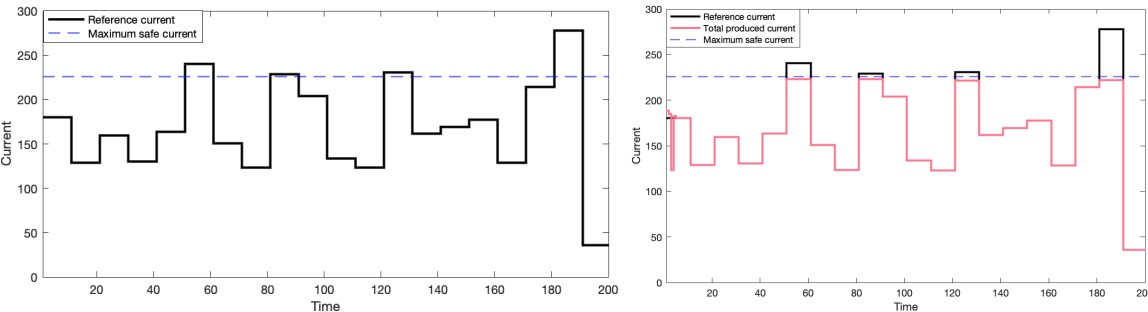

Figure 2: **(left)** Reference current to distribute over two electric motors. **(right)** The results of ARTEO for the given reference current.

where $x$ is the torque of the electric motors, $Cr_t$ is the given total current reference at time $t$, and $\mu_{TC_i}(x)$ and $\sigma_{TC_i}(x)$ represent the mean and standard deviation of GP regression for the unknown $p_{TC_i}$ of produced electric current for machine $i$ at torque $x_t$. $z$ is the hyperparameter for driving exploration. The operation range limit of torque is implemented as bound constraints

$$0 \leq x_i \leq 38 \text{ Nm}, \quad \forall i \in \mathcal{I}. \tag{10}$$

Lastly, the safety limit of the produced current for chosen electric motors is decided as 225.6 A according to the default value in the GEM environment and $g_t$ is formulated as

$$\sum_{i=1}^{2} \mu_{TC_i,t}(x) + \beta_t \sigma_{TC_i,t}(x) \leq 225.6 \text{ A} \tag{11}$$

where $\beta_t$ decides the width of the CI as in Equation (4). Computing an exact value for $\beta_t$ requires the knowledge of $B$, which is not known in practice. To overcome this issue in our experiments, we follow the empirical setup in previous work in safe BO (Berkenkamp et al., 2016; König et al., 2021) and directly use confidence bounds of GP with setting $\beta_t = 1.96$ for all $t = \{1, \ldots, T\}$.

After building the optimization problem, the reference current to follow by ARTEO is generated as in left in Figure 2. The reference trajectory is designed in such a way that it includes values impossible to reach without violating the safety limit (the black points over the blue dashed line). The aim of this case study is to follow the given reference currents by assigning the torques to the motors while the current to be produced at the decided torque is predicted by the GPs of each motor since the explicit form of the function that models produced current for given torque is an unknown black-box function. Therefore, ARTEO learns the torque-current black-box function first from the given safe seed for each motor, then updates the GPs of the corresponding motors with its noisy observations at each time step.

The safe seed of each motor consists of two safe points which are chosen from collected data in the GEM environment. The kernel functions of both GPs are chosen as a squared-exponential kernel with a length scale of 215 (set experimentally). The value of exploration parameter $z$ is decided as 25. Hyperparameter optimization techniques for $z$ are discussed and employed in this case study in Appendix C.1.3. The result of the simulation for the reference in left in Figure 2 is given right in Figure 2. Figure 2 shows that ARTEO is able to learn the torque-current relationship for given electric motors and optimize the torque values to produce given reference currents after a few time steps without violating the maximum safe current limit for this scenario.

When exploration starts (second time step) ARTEO prioritizes safe learning for the given $z$ value. In the second time step, ARTEO sends Machine-2 to a greater torque which helps decrease uncertainty while sending Machine-1 to a smaller torque to not violate the maximum safe current threshold. The comparison of estimated and real torque-current curves for the second and last time steps of the simulation is given in Figure 3. In this experiment, we keep $z$ constant as $z = 25$. The effect of the exploration hyperparameter, online hyperparameter optimization and recommended approaches to set it to an optimal value in an offline setting is discussed with additional experiments in Appendix C.1.

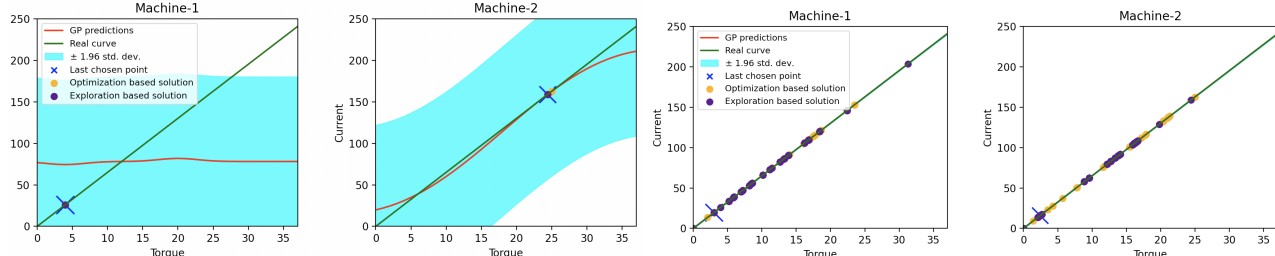

Figure 3: The second and last time steps of the simulation. The safe seed set includes two points for each electric motor within the operating interval. The blue-shaded area represents the uncertainty, which is high at the beginning due to unknown regions. GP-predicted torque-current lines are converged to the actual curves of each electric motor. Purple-coloured sample points are chosen by exploration.

### 4.1.2 Comparison with entirely GP-based safe exploration and optimization algorithms

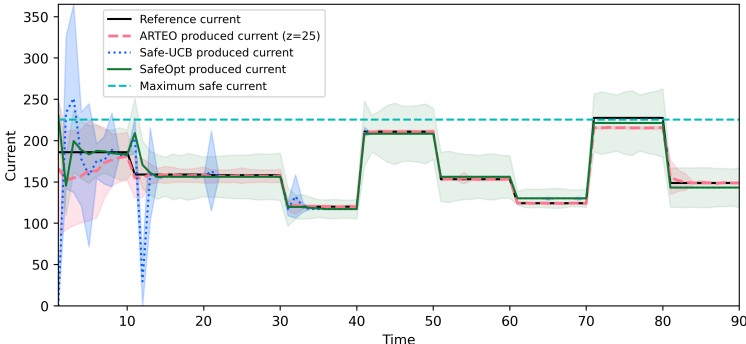

Figure 4: Comparison of produced currents of ARTEO and Safe-UCB algorithms. Shaded areas demonstrate ±1 standard deviation added version of the same-colour used algorithm.

We compare ARTEO with pure GP-based safe exploration and optimization algorithms. Our benchmark includes Safe-UCB, a safety-aware version of GP-UCB, and SafeOpt (Sui et al., 2015). We implement GP-UCB as in (Srinivas et al., 2010) to explore and exploit the decisions that minimize the objective function. Since the functional form is unknown in the original GP-UCB, we develop Safe-UCB without leveraging this information. A main difference between the original GP-UCB and Safe-UCB is that we add another GP to collect data of $g$ and learn the unsafe torque values by using the upper confidence bound as in ARTEO. The complete algorithm of Safe-UCB is given in Appendix C.2. We simulate the reference current value in Figure 4 with 50 different safe seeds. The results show that Safe-UCB tends to violate the safety constraint and operate the electric motors far from optimal values at first explored points due to high standard deviation in its predictions. Even though ARTEO is affected by the same level of uncertainty at the beginning of the simulation, the standard deviation of its decisions is significantly less than Safe-UCB and SafeOpt. A major concern for SafeOpt is that the choice of safe seeds affects its performance compared to ARTEO as evidenced by relatively high variance in its decisions at all iterations over different experiments with different safe seeds (green shaded area in Figure 4).

Moreover, Figure 4 demonstrates that while Safe-UCB and SafeOpt under or over-deliver produced current in several points, ARTEO is more capable of finding points that minimize the objective function unless the reference value is unsafe with respect to safety constraint. We further compare the cumulative regret of

algorithms as in Figure 1 by defining the regret $r_t$ at time step $t$,

$$r_t = |\max(Cr_t, 225.6) - \sum_{i=1}^{2} \mu_{TC_i,t}(x)|, \quad \forall t. \tag{12}$$

Figure 1 shows the superiority of ARTEO to learn the black-box functions in a more accurate manner and leverage them in optimization with suffering lower regret. Here, it is necessary to note that the effect of step size in the discretization of decision space substantially impacts SafeOpt's cumulative regret. It is theoretically possible to obtain better results for this case study with SafeOpt by decreasing the step size. However, the associated computational burden proves the impracticality of finer decision points. This drawback of SafeOpt can be seen right in Figure 1 and has been studied in (Berkenkamp et al., 2016). By incorporating the model structure into the decision-making process, our approach provides ARTEO with a significant advantage in terms of computational efficiency and achieving the minimum cumulative regret compared to pure GP-based safe exploration and optimization methods.

### 4.1.3 Online hyperparameter optimization

We use the total uncertainty $U_t$, which is quantified as in Equation (7), and instantaneous regret $r_t$, which is quantified as in Equation (12), to train $\mathcal{GP}_{hyp}$ and share the results in Figure 5. Compared to the simulation results in Figure 2 with constant $z$, $\mathcal{GP}_{hyp}$ set to $z$ values greater values at the start due to high uncertainty and inaccurate estimates of GP and begin to follow the reference signal successfully after few time steps. Both metrics work to set $z$ in an online manner, however, total uncertainty based $\mathcal{GP}_{hyp}$ performs better on a few occasions (zoomed in left Figure 5), can be seen in Figure 5.

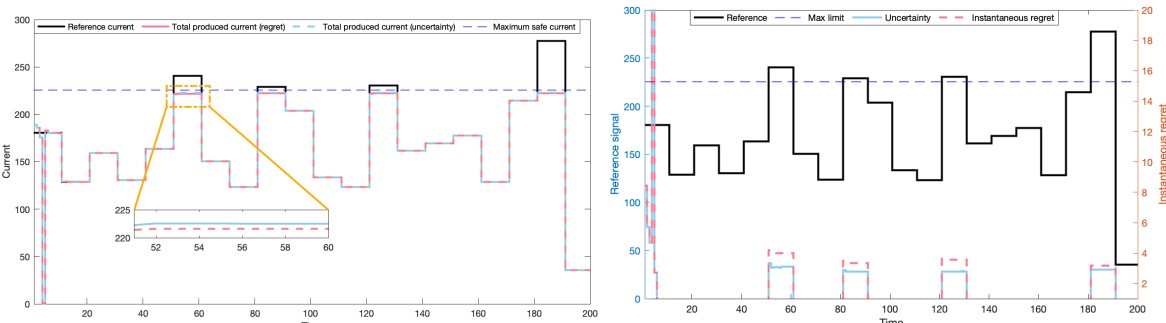

Figure 5: **(left)** The results of ARTEO for the given reference current when $z$ is decided by the lowest confidence bound of predictions of a GP which is trained by either (1) instantaneous regret and completed simulation step number or (2) total uncertainty and completed simulation step number. **(right)** Instantaneous regret comparison of total uncertainty-based and instantaneous regret-based online hyperparameter optimization approaches.

## 4.2 Online bid optimization

In the second experiment, we investigate the implementation of ARTEO in a multi-dimensional problem of online bid optimization from the advertiser perspective. In bid optimization, the advertiser sets bid values with the aim of achieving high volumes by maximizing the number of shown advertisements and high profitability by maximizing the return-on-investment (ROI) ratio. In most agreements, unsatisfied ROI causes financial losses for advertisers. It becomes more challenging to sustain high ROIs when the number of advertisements increases. Constraining the ROI to remain above a certain threshold is a common approach, however, this method does not guarantee to satisfy the ROI constraint with zero violation (Castiglioni et al., 2022). ROI is measured by the revenues and costs and revenue is unknown to the bidding algorithms which brings uncertainty to the online bid optimization problem. Therefore, safe optimization algorithms could be useful to set bid values under the uncertainty of the revenues.

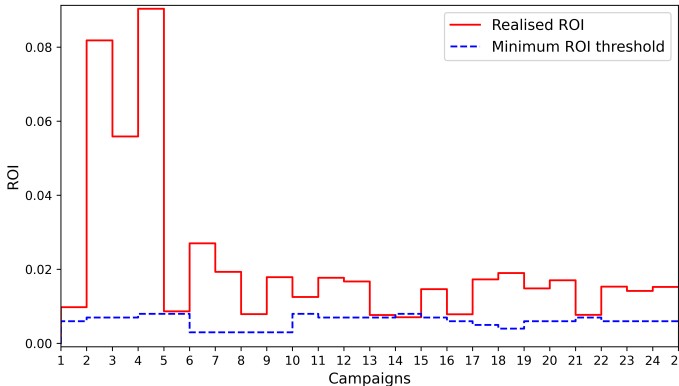

Figure 6: The minimum and achieved ROIs for campaigns. ARTEO is able to remain above the safety threshold.

We apply ARTEO to the iPinYou dataset (Liao et al., 2014). This dataset has been released by a leading DSP (Demand-Side Platform) in China and consists of relevant information for personalized ads such as creative metadata, interests of users, and advertisement slot properties with decided bid prices by their internal algorithm. It has been widely used as a benchmark to evaluate the performance of real-time-bidding algorithms (Zhang et al., 2016; Ren et al., 2017; Wang et al., 2017). We simulate our approach by creating different campaign subsets, each containing 20 ads. For each campaign subset $t$, we want to minimize the objective function

$$f_t(x) = \sum_{j=1}^{20} c x_j \mu_{C,t}(x_j) + \sum_{j=1}^{20} \sqrt{(x_j - \mu_{BP,t}(x_j))^2} - z \sum_{j=1}^{20} \left( \sigma_{C,t}(x_j) + \sigma_{BP,t}(x_j) \right) \tag{13}$$

where $j$ denotes the ad number in the campaign. The function $\mu_{C,t}(x_j)$ represents the mean prediction of a GP for the $j$th advertisement to get a click with set bid values $x_j$, and $\mu_{BP,t}(x_j)$ gives the mean prediction of a GP for the bid price of the $j$th ad in campaign $t$ based on previous observations. Given the vector of decision variables $x \in \mathbb{R}^{200}$, each element $x_j$ corresponds to the bid value for an advertisement in the campaign subset. The fixed budget constraint for the 200 ads in campaign $t$ is given by

$$\sum_{j=1}^{20} x_j \leq 180m. \tag{14}$$

The safe ROI constraint is constructed for the threshold $h_t$ for the campaign $t$ as follows

$$\frac{\sum_{j=1}^{20} \mu_{C,t}(x_j) - \beta_t \sigma_{C,t}(x_j)}{\sum_{j=1}^{20} x_j} \geq h_t. \tag{15}$$

Lastly, the bid values $x_j$ are bounded with non-negativity for all campaigns $t$ and for all advertisements $j$.

As opposed to the first experiment, where the changes in optimization goals were driven by changes in current references, an increase in $t$ in the online bid optimization example is driven by starting a new campaign. We construct two GPs in this experiment, the first GP learns the bid prices from past observations, and the second one models the impressions, which are represented in binary for clicks. The impressions are traditionally predicted by classifiers due to their binary representation. However, it is possible to cast it as a regression problem where we decide the binary representations after thresholding. Since we use covariance functions of GPs to model uncertainty, we cast it as a regression and guide our RTO with continuous values. The optimization algorithm bids an ad comparatively high when its value is higher than others.

Different feature sets in the dataset are used to compute posteriors based on relevance to the predictions. The GP of the bid price is initialized with the Matern kernel with $\nu = 1.5$ and is trained over 143 features whereas

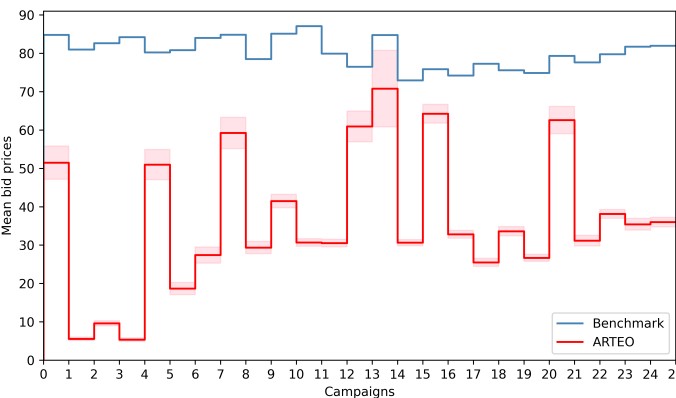

Figure 7: The mean bid prices for the given benchmark and ARTEO. ARTEO achieves higher ROI with lower costs. The shaded area represents $\pm 0.1$ standard deviation of optimized bid prices for the corresponding campaign.

the impression GP has the Squared Exponential kernel and 69 features. Further implementation details such as hyperparameters and safety limits could be found in Appendix C.3. The results of the simulation are given in Figure 6 and Figure 7. Our approach remains above the safety threshold while proposing lower bid prices compared to the given bid prices of the algorithm in Liao et al. (2014).

## 5  CONCLUSIONS

In conclusion, our work introduces the ARTEO algorithm, a novel approach for solving constrained stochastic optimization problems in safety-critical systems. By modelling the constraints and objectives as grey-box functions of decisions and incorporating Gaussian Processes (GPs) to capture uncertainty as explained in Section 2, we achieve improved performance compared to existing GP-based exploration and optimization techniques. Our mathematical programming framework, with constraints for safety, allows us to detect infeasibility with $1 - \delta$ failure probability and provides practicality in real-world scenarios. In Section 3, we presented that through an adaptive online optimization approach, ARTEO effectively explores the environment by making decisions at points of high uncertainty while maintaining a high probability of satisfying safety constraints. In Section 4, we demonstrated that by leveraging confidence bounds and empirical results, ARTEO yields safe and profitable outcomes in the problems we have investigated. To summarize, our contributions enhance the field of safe exploration and optimization, offering a valuable tool for addressing complex challenges in safety-critical systems. In future work, we aim to investigate the incorporation of prior knowledge for basis function selection in GPs, as well as substituting GPs with ensemble models to further enhance the capabilities of the ARTEO algorithm and expand its potential applications.

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

# A  Proofs

## A.1  Proof of Lemma 2.10

Let $\hat{p}^L = \mu_t(x) - \beta_t\sigma_t(x)$ and $\hat{p}^U = \mu_t(x) + \beta_t\sigma_t(x)$ where $\hat{p}^L \leq \hat{p}(x) \leq \hat{p}^U$. Given $g$ is monotonically related to $\hat{p}$, $g$ is a real-valued function $g(\Delta(x), \hat{p}(x))$ with a known explicit functional form of $\Delta(x)$, and $\hat{p}^L \leq \hat{p}(x) \leq \hat{p}^U$:

- if $g$ is monotonically related to $\hat{p} \Rightarrow g(\Delta(x), \hat{p}^L) \leq g(\Delta(x), \hat{p}(x)) \leq g(\Delta(x), \hat{p}^U)$.

- if $g$ is inversely monotonically related to $\hat{p} \Rightarrow g(\Delta(x), \hat{p}^U) \leq g(\Delta(x), \hat{p}(x)) \leq g(\Delta(x), \hat{p}^L)$.

*Proof.* Let $g$, $\Delta$ and $\hat{p}$ are continuous functions over domain $D$. We assume $g$ has a known functional form as defined by algebraic operations over known function $\Delta(x)$ and estimated function $\hat{p}(x)$ by using Gaussian processes to learn unknown function $p(x)$. It is given that $g(\cdot)$ is monotonically related to $\hat{p(\cdot)}$.

1. Consider the first case in Lemma 2.10 as for any $x, y \in D$ such that $\hat{p}(x) \leq \hat{p}(y) \Rightarrow g(\cdot, \hat{p}(x)) \leq g(\cdot, \hat{p}(y))$ (Definition 2.7). For chosen $x_1, x_2, x_3 \in D$ such that

$$x_1 \leq x_2 \leq x_3 \Rightarrow \hat{p}(x_1) \leq \hat{p}(x_2) \leq \hat{p}(x_3). \tag{16}$$

   With given $g$ is monotonically related to $p$:

$$\hat{p}(x_1) \leq \hat{p}(x_2) \leq \hat{p}(x_3) \Rightarrow g(\cdot, \hat{p}(x_1)) \leq g(\cdot, \hat{p}(x_2)) \leq g(\cdot, \hat{p}(x_3)). \tag{17}$$

   Let $\hat{p}^L = \hat{p}(x_1)$ and $\hat{p}^U = \hat{p}(x_3)$. By substituting terms in Equation (17), we obtain

$$\hat{p}^L \leq \hat{p}(x_2) \leq \hat{p}^U \Rightarrow g(\cdot, \hat{p}^L) \leq g(\cdot, \hat{p}(x_2)) \leq g(\cdot, \hat{p}^U). \tag{18}$$

   We can replace $x_2$ with any $x \in D$ that satisfies $\hat{p}^L \leq \hat{p}(x) \leq \hat{p}^U$. Thus, we have:

$$\hat{p}^L \leq \hat{p}(x) \leq \hat{p}^U \Rightarrow g(\cdot, \hat{p}^L) \leq g(\cdot, \hat{p}(x)) \leq g(\cdot, \hat{p}^U). \tag{19}$$

2. Consider the second case in Lemma 2.10 as for any $x, y \in D$ such that $\hat{p}(x) \leq \hat{p}(y) \Rightarrow g(\cdot, \hat{p}(x)) \geq g(\cdot, \hat{p}(y))$ (Definition 2.8). For chosen $x_1, x_2, x_3 \in D$ such that

$$x_1 \leq x_2 \leq x_3 \Rightarrow \hat{p}(x_1) \leq \hat{p}(x_2) \leq \hat{p}(x_3). \tag{20}$$

   With given $g$ is inversely monotonically related to $p$:

$$\hat{p}(x_1) \leq \hat{p}(x_2) \leq \hat{p}(x_3) \Rightarrow g(\cdot, \hat{p}(x_1)) \geq g(\cdot, \hat{p}(x_2)) \geq g(\cdot, \hat{p}(x_3)). \tag{21}$$

   Let $\hat{p}^L = \hat{p}(x_1)$ and $\hat{p}^U = \hat{p}(x_3)$. By substituting terms in Equation (21), we obtain

$$\hat{p}^L \leq \hat{p}(x_2) \leq \hat{p}^U \Rightarrow g(\cdot, \hat{p}^U) \leq g(\cdot, \hat{p}(x_2)) \leq g(\cdot, \hat{p}^L). \tag{22}$$

   We can replace $x_2$ with any $x \in D$ that satisfies $\hat{p}^L \leq \hat{p}(x) \leq \hat{p}^U$. Thus, we have:

$$\hat{p}^L \leq \hat{p}(x) \leq \hat{p}^U \Rightarrow g(\cdot, \hat{p}^U) \leq g(\cdot, \hat{p}(x)) \leq g(\cdot, p^L). \tag{23}$$

$\square$

## A.2 Proof of Theorem 2.11

Let Assumptions 2.1, 2.2, 2.3, 2.4, 2.5 and 2.9 hold. The maximum and minimum values of $g(\Delta(x), \hat{p}(x))$ lie on the upper and lower confidence bounds of the Gaussian process obtained for $\hat{p}(x)$ which are computed as in Lemma 2.10. For a chosen $\beta_t$ and allowed failure probability $\delta$ as in Equation (4),

- if $g$ is monotonically related to $\hat{p} \Rightarrow P\big[g_t(\Delta(x), \hat{p}^L) \leq g_t(\Delta(x), \hat{p}(x)) \leq g_t(\Delta(x), \hat{p}^U)\big] \geq 1 - \delta, \forall t \geq 1$

- if $g$ is inversely monotonically related to $\hat{p} \Rightarrow P\big[g_t(\Delta(x), \hat{p}^U) \leq g_t(\Delta(x), \hat{p}(x)) \leq g_t(\Delta(x), \hat{p}^L)\big] \geq 1 - \delta, \forall t \geq 1$

*Proof.* Theorem 2 by Chowdhury & Gopalan (2017) shows that the following holds with probability at least $1 - \delta$:

$$\forall t \geq 1 \ \forall x \in D, |\hat{p}(x) - \mu_{t-1}(x)| \leq \beta_t \sigma_{t-1}(x), \tag{24}$$

by choosing a $\beta_t$ as in Equation (4) under Assumption 2.3 and Assumption 2.5 (for proof, see Theorem 2 of (Chowdhury & Gopalan, 2017)). We can extract the inequality from absolute value as in the following

$$\mu_{t-1}(x) - \beta_t \sigma_{t-1}(x) \leq \hat{p}(x) \leq \mu_{t-1}(x) + \beta_t \sigma_{t-1}(x). \tag{25}$$

Define $\hat{p}^L$ and $\hat{p}^U$ as

$$\hat{p}^L = \mu_{t-1}(x) - \beta_t \sigma_{t-1}(x). \tag{26}$$

$$\hat{p}^U = \mu_{t-1}(x) + \beta_t \sigma_{t-1}(x). \tag{27}$$

Then, plug $\hat{p}^L$ and $\hat{p}^U$ into Equation (25) and obtain the following with $1 - \delta$ probability

$$\hat{p}^L \leq \hat{p}(x) \leq \hat{p}^U. \tag{28}$$

By the proof of Theorem 2.10, we can reflect this inequality to $g$ since $g$ is defined as monotonically related to $p$. Thus following statements hold with $1 - \delta$ probability,

- if $g$ is monotonically related to $p \Rightarrow g(\Delta(x), \hat{p}^L) \leq g(\Delta(x), \hat{p}(x)) \leq g(\Delta(x), \hat{p}^U)$,

- if $g$ is inversely monotonically related to $p \Rightarrow g(\Delta(x), \hat{p}^U) \leq g(\Delta(x), \hat{p}(x)) \leq g(\Delta(x), \hat{p}^L)$.

$\square$

# B Generalization to multiple safety constraints and black-box functions

We established the assumptions in Section 2.2 with the intention to clearly illustrate the fundamental principles of our approach, initially avoiding the added complexity of multiple constraints and variables. Here, we discuss the generalizability of our framework to scenarios involving multiple black-box functions ($|\mathcal{I}| > 1$) and multiple safety functions ($A > 1$).

In our proposed algorithm, each black-box function is modelled through a distinct GP, with regularity assumptions in Section 2.2 applicable to each function. Our framework accommodates multiple black-box functions by introducing a separate GP for each. This scenario is demonstrated in Section 4, where the first case study involves modelling the torque-current relationship of two different electric machines as two distinct black-box functions. Similarly, in the second experiment, we model the bid prices of advertisements and impression probabilities as separate black-box functions, each represented by its own GP. These examples underscore the adaptability of our framework to scenarios with multiple black-box functions.

Furthermore, Assumption 2.9 plays a crucial role in ensuring safety with high probability by enabling the propagation of uncertainty from $p_i, i \in \mathcal{I}$ to the evaluation of $g$, as elaborated in Theorem 2.11. In setups with multiple safety constraints, provided that the stated assumptions in Section 2.2 hold for each $g_{a,t}$ and $p_i$, the primary difference lies in addressing a multi-constraint problem. Modern solvers are well-equipped to handle optimization problems with multiple constraints, as long as the complexity does not render the

problem infeasible. Identifying specific conditions and interrelationships among safety constraints that might lead to infeasibility remains an open question for future research.

In conclusion, our methodology demonstrates flexibility and scalability, making it well-suited for a wide range of applications involving complex optimization scenarios with multiple unknowns and constraints.

## C  Experiment details

We have constrained nonlinear problems in the experiments section, and we choose interior-point and sequential-least square programming (SLSQP) algorithms to solve our first and second problems, respectively.

### C.1  Implementation of ARTEO to electric motor current optimization

#### C.1.1  Electric motor simulation environment details

Gym-Electric-Motor (GEM) is an environment that includes the simulation of different types of electric motors with adjustable parameters such as load, speed, current, torque, etc. to train reinforcement learning agents or build model predictive control solutions to control the current, torque or speed for a given reference. In electric motors, the operation range is limited by nominal values of variables to prevent motor damage. Furthermore, there are also safety limits for some parameters, such as the maximum safe current limit to avoid excessive heat generation.

Table 1: Electric motor parameters

| ELECTRIC MOTOR | $R_a$ | $L_a$ | $\psi_e$ | $J_{rotor}$ |
|---|---|---|---|---|
| MACHINE-1 | 0.016 | 1.9E-05 | 0.165 | 0.025 |
| MACHINE-2 | 0.01 | 1.5E-05 | 0.165 | 0.025 |

#### C.1.2  The impact of exploration hyperparameter

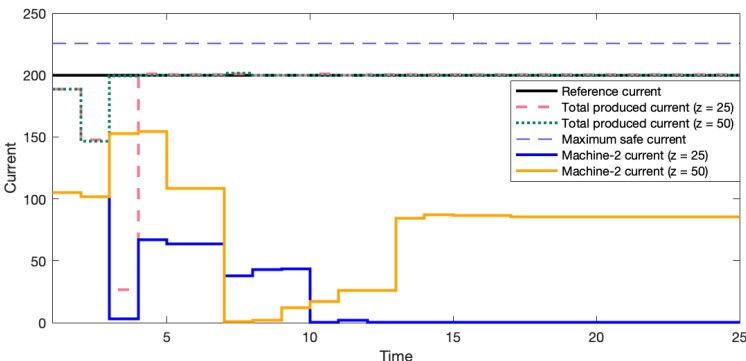

Figure 8: Greater $z$ values encourage exploration at points with a greater standard deviation.

The exploration in ARTEO is driven by the $z$ hyperparameter. It is possible to create different strategies according to the requirements of the problem by setting different values to this hyperparameter. To demonstrate this, we simulate a constant reference current scenario with different $z$ values, as in Figure 8. This figure exhibits that greater $z$ values lead to more frequent changes in reference torque values while preserving the ability to satisfy the reference current. It is expected that more frequent changes in the operating points assist in decreasing the total uncertainty in the environment faster. This is demonstrated in Figure 9 for different $z$ values for the reference scenario in Figure 8. Hyperparameter optimization methods could be leveraged to choose the value of $z$ to achieve minimum regret where regret at time $t$ is defined as Equation (12).

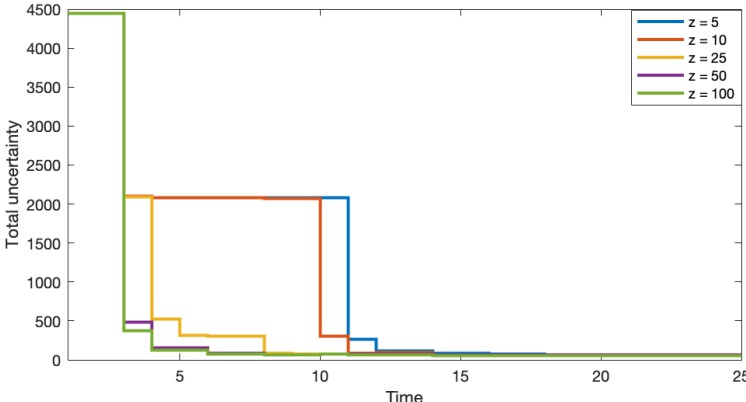

Figure 9: Total uncertainty decreases with a greater rate over time for greater $z$ values in an environment with a constant reference.

### C.1.3  Offline hyperparameter optimization for $z$

We analyse here the offline hyperparameter optimization approaches for ARTEO as opposed to our experiments to take advantage of offline data when it is available. We compare two hyperparameter optimization methods as the grid-search (LaValle et al., 2004) and Bayesian optimization (BO) (Frazier, 2018). We evaluate different $z$ values based on the cumulative regret at the end of the simulation of reference in Figure 10. For grid-search, we evaluate the cumulative regret with $z$ taking values of 5, 10, 25, 50, and 100. The results in Figure 11 show that the most suitable $z$ value for the given reference is 25 amongst the evaluated values.

As an alternative hyperparameter technique to grid-search, BO is also applied to the simulation of the given reference. BO is an optimization method that builds a surrogate model with evaluations at chosen points and then chooses the next value to be evaluated based on the minimization/maximization of the chosen acquisition function, which is specified as a lower confidence bound in our implementation. Further details of BO can be found in Frazier (2018). We limit the maximum number of evaluations with 35 points and the results in Figure 11 demonstrate that the surrogate model of BO suggests that the minimum cumulative regret at the end of the simulation (of Figure 10) when $z = 28$.

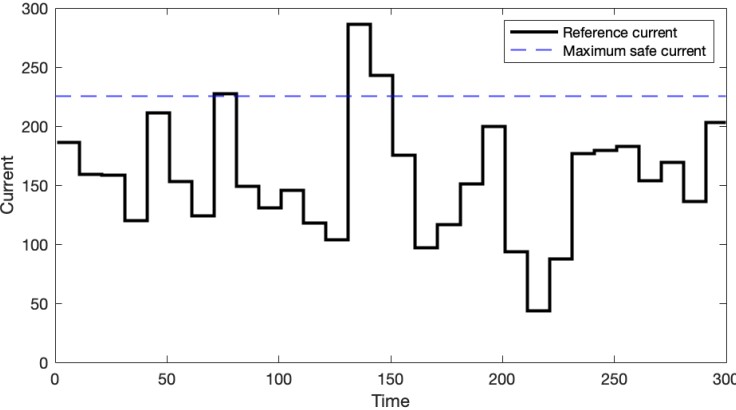

Figure 10: Reference current of the hyperparameter optimization simulation (longer simulation of Figure 4).

### C.2  Implementation of Safe-UCB to electric motor current optimization

In electric motor current optimization experiments, $f$ is calculated as the difference between the given reference current and the total produced current. Thus, we search points that minimize the value of $f$ which

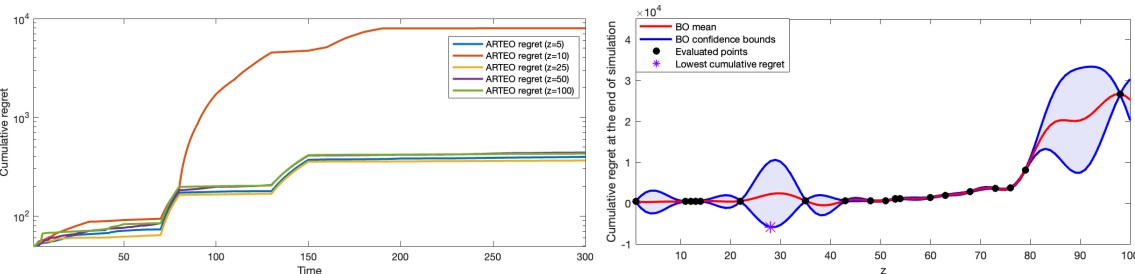

Figure 11: Hyperparameter optimization results for $z$ in electric motor current optimization with reference scenario in Figure 10. The left figure depicts the cumulative regret (in log scale) over time for each evaluated $z$ value in grid-search. The right figure shows the evaluated $z$ values in BO with lower confidence bound acquisition function and fit GP model mean and confidence bounds.

leads us to use lower confidence bound of $f$ by following the optimism principle of Lai & Robbins (1985). The safety function $g$ is defined as the difference between the safety limit value $h = 225.6$ and the total produced current. Hence, the chosen points are safe as long as the $g$ value of chosen points remains above zero.

---

**Algorithm 2** Safe-UCB

---

**Input:** Decision set $D$ for each variable $i \in \{1, .., n\}$, GP priors for $GP^f$ and $GP^g$, safe seed sets $S_0^f$ and $S_0^g$
**for** $t = 1, ..., T$ **do**
    Update $GP^f$ by conditioning on $S_{t-1}^f$
    Update $GP^g$ by conditioning on $S_{t-1}^g$
    Choose $x_t^* = \arg\min_{x \in D} \mu_t^f(x) - \beta_t \sigma_t^f(x)$ subject to $\mu_t^g(x) - \beta_t \sigma_t^g(x) \geq 0$
    $y_t^f \leftarrow f(x^*) + \epsilon_t^f$
    $y_t^g \leftarrow g(x^*) + \epsilon_t^g$
    $S_t^f \leftarrow S_{t-1}^f \cup \left\{ x^* : y_t^f \right\}$
    $S_t^g \leftarrow S_{t-1}^g \cup \left\{ x^* : y_t^g \right\}$
**end for**

---

### C.3 Environment details for online bid optimization

The GP of the bid price is initialized with the Matern kernel with $\nu = 1.5$ and is trained over 143 features whereas the impression GP has the Squared Exponential kernel and 69 features. The safe seeds start with 30 samples, which is higher than our first experiment since this is a higher dimensional problem. As a minimum ROI threshold, 90% of the given benchmark data ROI is set due to having a strict budget and ROI requirements in our setup. We partition the selected subset of the dataset into 25 campaigns. Each campaign has its ROI threshold and budget, which are calculated as Equation (14) and Equation (15).

The algorithm starts with a safe seed set to compute the posteriors of GPs, and then for each campaign, it utilizes the mean and standard deviation of GP posteriors to measure ROI and click probability. During the RTO phase, the higher bid prices for higher estimated click values are encouraged within a fixed budget, and the difference between the predicted bid price by GP and the proposed bid price by RTO for each advertisement is accumulated and introduced as a penalty in the objective function. Thus, the algorithm does not put the entire budget into the highest-valued ad within the campaign. At the end of each campaign, true bid prices and clicks with additive Gaussian noise are used to update the posteriors of GPs. The feedback is given only for ads in the campaign with a non-negative optimized bid price which leads to high standard deviations for non-bid similar ads. The environment becomes available for exploration after spending less than the sum of predicted bid prices and satisfying minimum thresholds in two consecutive campaigns. Hence, $z$ is set by $\mathcal{GP}_{hyp}$ to excite the RTO to make decisions at points that could reduce uncertainty in predictions.

