# OpenReview forum: "Adaptive Decision-Making for Optimization of Safety-Critical Systems: The ARTEO Algorithm"
_TMLR — Rejected by TMLR_

### Review · Reviewer_AWtR · 2023-06-14

**Summary Of Contributions:**

The paper addresses the decision-making problem for safety-critical systems. The authors develop ARTEO, an algorithm based on Gaussian Processes (GPs) that, besides optimizing for the best decision under uncertainty, accounts for the presence of safety constraints. In this way, GPs incorporate the uncertainty on both the objective function and on the constraints. Under specific assumptions related to the relative monotonicity between the objective function and the constraints, the paper provides confidence bounds on the error due to the noise. An empirical validation is conducted on an eclectic motor current optimization problem and on an online bid optimization problem.

**Audience:**

No

**Broader Impact Concerns:**

No ethical implications/concerns identified.

**Claims And Evidence:**

Yes

**Requested Changes:**

Please refer to the Weaknesses.

**Strengths And Weaknesses:**

***Strengths***
1. Addressing a relevant problem from the practical application perspective, i.e., incorporating uncertain constraints in the decision/optimization process.
2. An experimental validation that illustrates some of the aspects of the proposed approach.

***Weaknesses***
1. [Formulation] In Section 2, the paper provides the problem formulation by introducing a cost function $f$ which depends on the decision $x_t$ and on the "system characteristics" $v_t$. It is unclear to me the actual meaning of this latter term. Specifically, while I understand (from an intuitive perspective) that the effect of a decision depends on the features of the system, I do not get why it should depend on time. Wouldn't it be equivalent to consider a non-stationary cost function $f_t$? Moreover, the dependence on $\lambda$ is unclear to me. What is the meaning of $\lambda$? Furthermore, the interplay between the decision $x_t$ and the system characteristics $v_t^\lambda = p_{\lambda}(x_t)$ should be explained in more detail. Overall, at this point, it seems to me that the formulation is unnatural and complex.

2. [Regularity Assumptions] First of all, it is not fully explained what "the cost function $f$ and safety function $g$ might include known terms $\Delta$" means. How the $\Delta$s affect those functions and how this knowledge helps the optimization procedure? Furthermore, the (inverse) monotonically related assumptions seem very restrictive. The authors acknowledge this in Section 3.4. Nevertheless, to convince the reader that those assumptions are reasonable, some examples of practical applications in which they are satisfied should be presented. This is particularly important since those assumptions make the proof of Lemma 2.4 and Theorem 2.5 (the main theoretical contributions of the paper) quite straightforward.

3. [Missing Regret/Sample Complexity Guarantees] The theoretical contribution of the paper limits to Lemma 2.4 and Theorem 2.5, that as already mentioned become straightforward, given the (inverse) monotonically related assumptions. The paper completely misses a theoretical analysis to characterize the full execution of the algorithm in terms of regret and sample complexity guarantees. The authors should elaborate on why it is missing or why it is not necessary.

***Minor Issues***
- The numbers/writings in the plots are very small
- Missing punctuation in equations

---

> ### Author Response · Authors · 2023-10-27
>
> Dear Reviewer AWtR,
>
> Thank you for your time and effort in reviewing our work. We provide detailed clarification to address the issues raised in your comments below:
>
> - **Formulation:** We have made a major revision in our formulation as suggested by reviews and addressed the concerns of the reviewer by simplifying the formulation (in sections 2 and 3). We thank the reviewer for suggesting to use of a non-stationary cost function $f_t$ for a better representation of our problem. In the current version, we implemented this feedback.
> - **Regularity Assumptions:** The approach in our paper is to construct the safety constraint from as much prior knowledge as possible (based on physics and other domain knowledge, represented as $\Delta$) and then use data-driven models (GPs) to learn parts of these functions for which we don’t have as much information. The concept of monotonicity pertains to how the data-driven parts (GPs) relate to the safety constraint. We further revised our regularity assumptions sections and motivated how to know $\Delta$ would help in this optimization problem by demonstrating our formulation incorporating prior knowledge outperforms entirely black-box-based BO methods (subsection 4.1.2) and referring to prior work with findings including prior knowledge leads improved regret [1].
>
>     For the monotonicity assumption, we updated the limitations subsection and referred to case studies dealing with monotonic constraints as follows:
>  >ARTEO's theoretical safety guarantee is based on the assumption of monotonicity between the safety constraint and black-box functions. However, it's worth noting that monotonic relationships in functions are not uncommon in practical settings. For instance, \citet{arellano2009chance} demonstrates applications from the process industry which have safety constraint functions that are strictly monotonic with respect to some variables. Furthermore, \citet{10.5555/2946645.3007062} investigates methods to learn models where the data exhibits monotonic relationships. Lastly, \citet{nieves2022semi} models a case study where the constraints are monotonic and surveys other works concerned with monotonic constraints and functions. These references, along with our own case studies, serve to demonstrate that such monotonic relationships are not just theoretical constructs but are encountered in actual, practical scenarios. While these are just a handful of examples, they underscore the existence of this relationship in diverse settings.
> - **Missing Regret/Sample Complexity Guarantees:** We thank the reviewer for this comment. We agree that providing regret and sample complexity analysis would be an interesting addition to the paper. However, the proposed algorithm works with an extended objective function $f_t$ composed of the original cost $C_t(\cdot)$ and the uncertainty quantification $U_t(x)$, scaled by the hyperparameter $z$. Thus, the solution of the optimization problem with $f_t(\cdot)$ will depend on the numerical solver. The impact of the solver and its numerical properties on the solution is an addition planned for future work. For now, we have provided the empirical regret analysis for our first case study in the paper (Figure 1 and subsection 4.1.2).
>
> If you have any additional questions or comments, we would be happy to have further discussions.
>
> Kind regards,
>
> Authors of Paper1180
>
>
> [1] Raul Astudillo and Peter Frazier. Bayesian optimization of composite functions. In International Conference on Machine Learning, pp. 354–363. PMLR, 2019

---

> > ### Comment · Reviewer_AWtR · 2023-11-12
> > **Re: Official Comment by Authors**
> >
> > I thank the authors for addressing my concerns.

---

### Review · Reviewer_6gww · 2023-07-31

**Summary Of Contributions:**

In the present study, the authors primarily aim to address an online decision-making problem constrained by safety measures. The problem under investigation involves a system characterized by substantial uncertainty, a result of Gaussian noise and an unknown functional form in both the objective and constraints. To establish a safe online control scheme in this context, the authors introduce an Adaptive Real-Time Exploration and Optimization (ARTEO) algorithm. This algorithm operates on key assumptions about the monotonic and inverse monotonic relationships between constraint functions and predicted system characteristics. Utilizing the uncertainty estimates derived from the learned Gaussian process (GP) regression models, ARTEO develops a confidence interval that ensures adherence to safety constraints. Importantly, these uncertainty estimates dynamically guide environmental exploration, fostering more effective online decision-making. Empirical results derived from tasks related to electric motor current optimization and online bid optimization illustrate ARTEO's capabilities. The algorithm effectively estimates partially uncertain constraints and objectives, optimizing decisions to yield safe and profitable outcomes.

**Audience:**

Yes

**Broader Impact Concerns:**

None.

**Claims And Evidence:**

Yes

**Requested Changes:**

1. The motivation behind the main theoretical results needs to be more clearly articulated.

2. Including empirical results from more complex environments would significantly bolster the robustness and generalizability of your findings.

3. It would be beneficial to thoroughly explain why modern machine learning methods, such as uncertainty-aware Reinforcement Learning or constrained decision transformers, are not suitable solutions for the online decision-making problem with uncertainty and constraints.

**Strengths And Weaknesses:**

**Strengths:**

1. The manuscript is clear and well-structured. The introduction of the problem is comprehensive, with sufficient technical details that substantially enhance the reader's comprehension.

2. Compared to other constrained optimization algorithms such as Safe-UCB (a safety-aware variant of GP-UCB) and SafeOpt, the ARTEO algorithm showcases commendable empirical performance.

**Weaknesses:**

1. **Rationality behind Theoretical Results**: The primary theoretical result of this paper lies in the development of confidence bounds on predicting system characteristics (Section 2.3). The proofs are intuitive, drawing significantly from existing results and key assumptions. However, how well these bounds facilitate constraint satisfaction in the constrained optimization problem (Formula 1) is not explicitly clear. The link between Theorem 2.5 and the main objective of the paper remains ambiguous. Importantly, there seems to be an oversight: all the bounds presented are upper bounds, namely, $P(...)\leq 1-\delta$. It must be noted that $P(...)=0$ would also satisfy these bounds, irrespective of the magnitude of $\delta$. Arguably, a lower bound on a desired event would be more meaningful.

2. **Novelty**: To my understanding, the application of Gaussian Processes (GP) to estimate upper confidence bounds, though intuitive, is not a novel concept. The primary contribution of this paper is the adaptive and adjustable parameter $z$. While the adjustment method for $z$ in Algorithm 1 is indeed intuitive, the paper falls short of thoroughly exploring its role in facilitating more stable control in safety-critical systems.

3. **Limited Generalizability**: The empirical results of this paper are based on two simple environments (an electric motor current optimization and online bid optimization) in a low-dimensional state space. Why these evaluation benchmarks challenging are is not clearly articulated. Furthermore, it remains uncertain how these results could be generalized to more complex real-world problems, such as autonomous driving or robotic control.

Minor.

1. Page 5, Theorem 2.4 -> Lemma 2.4
2. It seems that algorithm 1 has more than one unknown.
3. Algorithm 1, line 10, wrong notation. should be: $\forall{i \in [1,...,n]}, x_i=\arg\min_{x_i \in D_t} f s.t. g$.
4. Algorithm 1, line 11, it seems $p_\lambda$ is already known.

---

> ### Author Response · Authors · 2023-10-27
>
> Dear Reviewer 6gww,
>
> Thank you for your time and effort in reviewing our work. We provide detailed clarification to address the issues raised in your comments below:
>
> - **Rationality behind Theoretical Results:**  Thank you for highlighting the need for a clearer connection between the developed confidence bounds and their role in facilitating constraint satisfaction.
>
>     In the updated manuscript, Assumption 2.2 states that we know the functional forms of both $f(\cdot)$ and $g$, while the functional form of $p_i(\cdot)$ remains unknown. This addition emphasizes the challenge we aimed to address: ensuring $g(\Delta(x), p_i(x)) < h$ with high probability. As you rightly pointed out, achieving this requires propagating the uncertainty about $p$ to $g$. This propagation is detailed in Lemma 2.10, which we have referenced more explicitly to clarify the relationship. We believe that this amendment provides a clearer understanding of our problem formulation and the significance of the derived confidence bounds in ensuring constraint satisfaction.
>
>     We are grateful for your keen observation regarding the oversight in our bounds. You are correct in noting that the presented bounds are upper bounds, and the way they were initially formulated could indeed be satisfied even if \(P(...) = 0\). We have rectified the signs in equation (3) and the associated theorem. The correct formulation should indeed present a lower bound on the desired event, ensuring that we are within the bounds with a probability of at least \(1-\delta\). We have made the necessary corrections in the manuscript and have ensured that the proofs and subsequent derivations are consistent with this change.
> - **Novelty:** Thank you for your feedback on novelty. We revised our contributions list at the end of the Introduction section and we value this opportunity to clarify and elaborate on our work further.
>
>     Our problem formulation distinguishes from prior work on GP-based safe optimization (safe BO) by modelling our objective function $f$ and $g$ through both known first-principle-driven functions $\Delta(\cdot)$ and unknown black-box functions $p(\cdot)$. In this setting, in order to extend safety guarantees of black-box optimization through propagating the confidence bounds of $p$ to $g$, we need to establish a relationship between $p$ and $g$. To the best of our knowledge, our paper is the first work that assumes a certain (monotonicity) relationship between an unknown function ($p$) and a safety constraint ($g$) represented as a nested function $g(\Delta(x)$, $p(x)$ of this unknown function ($p(x)$) with other elements (known functions, $(\Delta(x))$ in the context of the given task to establish safe optimization. Furthermore, previous work [1] and our comparison with established safe BO methods such as SafeOpt and Safe-UCB in the paper demonstrate that the optimization benefits with lower cumulative regret from incorporating prior knowledge into the algorithm when the objective and safety functions are formed as grey-box models.
> - **Limited Generalizability:** We thank the reviewer for the comment highlighting the weaknesses of the description of the case studies. We have now added the detailed descriptions of the challenges to the paper. The first case study, on current optimization in electric motors, was chosen to highlight that ARTEO preserves safety. To focus on safety, the dimensionality of the problem was kept low (2 variables/1-constraint) to illustrate the idea of ARTEO.
>
>     The second case study, online bid optimization, was chosen to demonstrate the performance of ARTEO in a high-dimensional case study. The challenges in this example are: (i) the use of real data for bid optimization from the released source, (ii) high dimensionality with 200 decision variables where each variable is the bid price of an advertisement in a campaign and one campaign consists of 200 advertisements, (iii) presence of binary variables in the optimization problem increasing the uncertainty induced by continuous reformulations.
>
>     The ARTEO algorithm has been developed for online optimization of problems where direct measurements or deterministic models are unavailable. As a result, the algorithm is application-agnostic and can be applied to solving real-world problems in a similar way as SafeOpt and Safe-UCB [2,3].
>
> If you have any additional questions or comments, we would be happy to further discuss.
>
> Kind regards,
>
> Authors of Paper1180
>
> [1] Raul Astudillo and Peter Frazier. Bayesian optimization of composite functions. In International Conference on Machine Learning, pp. 354–363. PMLR, 2019
>
> [2] Somil Bansal and Claire J Tomlin. Control and safety of autonomous vehicles with learning-enabled components. Safe, Autonomous and Intelligent Vehicles, pp. 57–75, 2019
>
> [3] Alexander Wischnewski, Johannes Betz, and Boris Lohmann. A model-free algorithm to safely approach the handling limit of an autonomous racecar. In 2019 IEEE CCVE

---

> > ### Comment · Reviewer_6gww · 2023-11-19
> >
> > Thank the author(s) for responding to my concern and addressing the issue. After reading the updated paper and comments from other reviewers, I have no further concerns, although I still believe the novelty and empirical study are not very strong.

---

### Review · Reviewer_JoQ7 · 2023-10-13

**Summary Of Contributions:**

This manuscript addresses optimization of a partially known objective function under safety constraints. The problem resembles the typical problem considered in safe Bayesian optimization (Safe BO), yet assumes a nested structure for the objective function like $f(x, g(x))$, where f(.) is known, and g(x) modeled with a GP. For this problem, an algorithm (ARTEO) is presented with statements about safety guarantees. The algorithms is illustrated in two numerical examples.

**Audience:**

Yes

**Broader Impact Concerns:**

/

**Claims And Evidence:**

No

**Requested Changes:**

Please see the mentioned weaknesses.

Given the scope of the weaknesses, in my evaluation, this manuscript is not suitable for TMLR in its present form.  In my evaluation, the manuscript needs a complete revision with regards to the results, the presentation, and the positioning wrt the state of the art.

**Strengths And Weaknesses:**

## Strengths

(S1) Data-driven optimization under safety constraints is a relevant problem, both from methodological, as well as practical point of view.

(S2) Leveraging prior knowledge about the problem structure is useful.

(S3) Code for the algorithms and numerical examples is provided.



## Weaknesses

(W1) A key assumption in this work is that a bound $B$ for the RKHS norm is known. This is a strong assumption (albeit common in Safe BO literature), which should be stated as such.

In fact, the authors do not provide any reasoning of how they can find $B$ in their case studies (they state that they "choose a large $B$"), nor do they state the actual numbers in the case studies.  I would argue that without any justification of the choice of $B$ (why is that number for $B$ a reasonable choice in the particular application?), one cannot claim that the results in the case studies are safe.  Hence, IMO, it is not justified to call the algorithm in these applications "safe". "Cautious" is a better and more honest word.  As this work is advertised as "safe optimization", I see this as a major problem in the presentation.  At the very least, this limitation should be clearly stated (including in the Limitations section (Sec. 3.4)), and how B is obtained in the examples, should be discussed.

(W2) The problem that is considered is very close to Safe BO, which has been well-studied in recent years. However, Safe BO is not properly (re-)presented in the discussion of the related work, and from the introduction it does not become clear that the authors actually tackle a special case of that problem. In particular:

* At the end of the 2nd paragraph of the Introduction, it is stated about the aforementioned works: "they do not consider safety-critical constraints."  This is correct.  However, it sounds like that there is no such work that considered safety with BO or bandit optimization, which is misleading.
* In a similar vein, the next paragraph (first paragraph on page 2) introduced the problem considered in Safe BO ("modelled as safety constraints in optimization"), yet neither the term Safe BO, nor existing works are mentioned here. The way it is described, one might thing that this is a new problem setting.  However, this setting is not new, but exactly the setting of Safe BO, which has been proposed by Sui et al 2015.   The reference Sui 2015 needs to be cited as the one that introduces this problem.  (In the next paragraph, the reference is given, but this should come earlier.)
* Given the close relation, there should be a discussion on other, recent Safe BO literature. Some recent works include Kirschner (2019), Sarikhani (2021), Sukhija (2023). I suggest to explicitly discuss how the problem and approach of this work relates to existing approaches in Safe BO.
* In the current discussion of related work (2nd paragraph on page 2), the existing work is contrasted with the statement "or apply a trade-off strategy between optimal decisions and exploration."  -- It is not clear to me, what the authors mean and how this is a contrast because there is - conceptually - always such a trade-off between "optimal decisions" (which I interpret as exploitation) and exploration.
* Later in the experimental section, the authors do compare to SafeOpt.  The problem considered in this paper thus seems to actually be a special case of the Safe BO problem.  Thus, it should be stated as such.
* In Sec. 2, please discuss how the problem (1) relates to and/or is different from optimization problems addressed in Safe BO literature.

(W3) Similarly (but less severe), the introduction motivates the need for "exploration in an adaptive manner".  However, no related work on adaptive/time-varying BO and maybe other approaches is discussed.

(W4) The bullet point list of contributions is helpful.  However, I suggest to make the contributions more specific.  For example, the sentence

"We capture uncertainty via GPs and use the estimates from the GP in a mathematical programming framework to find decisions that satisfy the constraints with high probability while maximizing the objective."

is misleading, because what it describes has been done before (e.g., Safe BO), and is thus no contribution.

Further, in the sentence

"the mathematical programming formulation with hard constraints for safety provides a practical means to detect infeasibility"

I do not understand what the authors mean by this contribution. (also see my comment (W8) below on "hard constraints")

(W5) Notation and technical problem formulation is partly imprecise or sloppy, which makes it hard to clearly identify and appreciate the technical contribution. For example, in Sec. 2:

* The notation for \Lambda is confusing.  Is \Lambda just the set of integers from 1 to d?
* Why do the authors first introduce v_t, to then introduce v_t^\lambda?  Also, it is not clearly specified that \lambda is a super-index here, rather than the exponent.
* epsilon does not have an index t, which would mean that it is sampled only once (rather than at every time t).  I don't think this is what the authors mean.  Further, please state any independence assumptions on the noise (if the case).
* For the constraints, please specify whether g_a, or h_a, or both are known/unknown.
* (1) is stated as a static optimization problem (which is solved at time t, for the variable x_t).  It is not clear from this formulation, what role the time sequence introduced earlier (t=1, ..., T) plays.  Presumably the authors mean that the optimization problem is solved sequentially, so each evaluation x_t is then used to update the GP estimates and thus influences the optimization at time t+1.  Further, from what I understood earlier, the constraints are supposed to be satisfied for all t.  However, these aspects are not stated nor explained in the problem statement, which is thus incomplete.
* The minimization in (1) should be over x_t, not X.
* The description of the relations between GPs, RKHS, and confidence bounds is very sloppy.
* x_{t \lambda} in (10) is not clear / introduced.

(W6) It is proposed to estimate p from data using GP regression.  Please specify what data is needed and available for this.  From my understanding, you need data of the form (x_t, v_t) to estimate p via regression.  However, this would mean that v_t can be measured, which is missing in the problem statement.  Further, is such assumption realistic in your use cases?

(W7) Further assumptions (in addition to (W2)): Sec. 2.2, the sentence "We do not have any prior knowledge (...) to provide safety (...) we need to make some assumptions." is misleading.  Do you make assumptions (and thus need prior knowledge), not not?  I suggest to clearly state in one place what assumptions are needed.

Similarly, I don't understand the statement "We sample unknown characteristics p_hat from a GP with a positive definite kernel".  Do the authors mean that they assume that p_hat is sampled from a GP with the kernel that they assume as prior knowledge?

(W8) It is stated (e.g., in Sec. 2.3, but also elsewhere) that constraints are treated as "hard constraints" in the optimization problem.  However, parts of the constraint function are estimated with a GP, for which then confidence bounds are employed.  Hence, the constraint should hold only with some probability.  I don't understand how this can then be guaranteed as a hard constraint.  Instead, I would expect satisfaction of the constraint with some probability.

(W9) Sec. 2.3: In recent work, there are tighter confidence bounds available for GP regression by Fiedler et al (2021), which should be useful here and be mentioned.  See: https://arxiv.org/abs/2105.02796

(W10) From the description that is given in Sec. 4.1, I do not understand the case study, in particular, the optimization problem.  ("Learn the relationship between torque and current" -- what precisely is the optimization problem?)

(W11) Given the number of weaknesses mentioned above, the paper is overall difficult to follow.



### Minor

* "the cost function f and safety function g might include known terms \Delta" -- this is unclear.  What mathematical objects are these "known terms"?
* Lemma 2.1 has no proof.  (but is an adapted version of some other work)
* Are definitions 2.2 and 2.3 novel, or taken from existing work.  Please state this, or add references.
* The claimed safety guarantee for the ARTEO algorithm (Theorem 2.5) is given in Sec. 2, where the algorithm has not even been specified (which happens in Sec. 3).
* Sec. 4.1.1: What do the authors mean by "set experimentally" when selecting the length scale in the motor example?
* Comparison to SafeOpt: "A major concern for SafeOpt is that its performance is highly affected by safe seed choice." -- I don't understand this sentence, and the comparison.  ARTEO also relies on a safe seed, no?
* The figures are poorly formatted.  In particular, the font size is too small.

### References
Yanan Sui, Alkis Gotovos, Joel W. Burdick, and Andreas Krause. Safe exploration for optimization with
Gaussian processes. 32nd International Conference on Machine Learning, ICML 2015, 2:997–1005, 2015.

Parisa Sarikhani, Benjamin Ferleger, Jeffrey Herron, Babak Mahmoudi, and Svjetlana Miocinovic. Automated deep brain stimulation programing with safety constraints for tremor suppression. Brain Stimulation, 14(6):1699–1700, 2021.

Johannes Kirschner, Mojmír Mutný, Nicole Hiller, Rasmus Ischebeck, and Andreas Krause. Adaptive and safe Bayesian optimization in high dimensions via one-dimensional subspaces. 36th International Conference on Machine Learning, ICML 2019, 2019-June:5959–5971, 2019.

B Sukhija, M Turchetta, D Lindner, A Krause, S Trimpe, D Baumann, GoSafeOpt: Scalable safe exploration for global optimization of dynamical systems, Artificial Intelligence 320, 103922, 2023.

C Fiedler, CW Scherer, S Trimpe; Practical and rigorous uncertainty bounds for gaussian process regression; Proceedings of the AAAI conference on artificial intelligence 35 (8), 7439-7447

---

> ### Author Response · Authors · 2023-10-27
>
> Dear Reviewer JoQ7,
>
> We are deeply thankful for the significant time and knowledge you have generously shared in evaluating our work through your detailed feedback! We addressed the mentioned weaknesses with major revisions in sections 1,2 and 3 and minor comments in section 4. Our detailed actions and comments for each mentioned weakness are as follows:
>
> - **W1:** We updated the limitations section by emphasizing this assumption is strong and common in safe BO literature and how we are building our confidence interval without knowing the value $B$.
> - **W2:** We revised the Introduction with the feedback related to introducing BO and safe BO and referring to the mentioned recent works. At the end of the Introduction and at the ARTEO algorithm section, we elaborated on how our problem formulation positioned in safe BO, TVBO and constrained real-time optimization.
> - **W3:** We discuss the time-varied BO and state our differences in subsection 3.3 of the revised version.
> - **W4:** We are grateful for this comment since it made us create a more specific contribution list and helped us to showcase the merits of our work better. Please see the end of the Introduction for the revised contributions list.
> - **W5:** We have done a major revision on the notation and problem formulation. We hope the current simplified version addresses the concerns of the reviewer.
> - **W6:** As stated by the reviewer, $v_t$ is not measurable. We clarify that we obtain observations through an oracle in the revised version. We updated the pseudocode accordingly.
> - **W7:** We state each assumption individually in the revised version under regularity assumptions which should address the concerns in this comment regarding the confusion about what are our assumptions.
> - **W8:** We apologise for the confusing statement earlier and we removed the mentioned sentence from the paper. To explain our perspective, our approach can be seen as happening in two steps – the problem formulation step including the learning of the unknown functions – and the optimization step where we numerically try to find a solution. The hard constraint comes into the picture for this second step. We use a nonlinear programming solver for the optimization step – and in these solvers, there is a distinction between the objective function and the constraints. From a problem formulation perspective, the constraints can be in the form of a penalty as part of the objective function (these would be soft constraints). This is not what we do in the optimization step - we provide the safety-related constraints as constraint functions which makes them hard constraints from the solver’s perspective. If the solver cannot find a solution that can satisfy these constraints the problem will be considered infeasible (without considering the objective function value). This information is valuable because it will indicate that with the probability level, we specify (for the constraint function calculation) for the satisfaction of the constraint no solution can be found.
> - **W9:** Thank you for sharing this study with us. We mentioned the tighter confidence bound finding in subsection 2.3 of the updated manuscript.
> - **W10:** In the revised version, we elaborate on the problem in the first case study, what is unknown and what GPs are modelling. We hope the revised version makes the optimization problem easier to understand for readers.
>
> If you have any additional questions or comments, we would be happy to have further discussions.
>
> Kind regards,
>
> Authors of Paper1180

---

> > ### Comment · Reviewer_JoQ7 · 2023-11-11
> > **Comments on authors' revision and discussion**
> >
> > I appreciate the authors' response, and I am very glad to hear that the authors found my comments helpful.  I am ok with most answer to my comments (albeit some were hard to verify, see below). Nonetheless, I have some remaining and additional concerns. Overall, I believe that the manuscript still needs a revision.
> >
> > My main concerns based on the changes in the manuscript:
> >
> > 1) **Problem formulation:** Before equation (1), C is introduced as a function R^n -> R, without time index.  In (1), C_t has the time index t, thus is an additional function of time.  However, it is unclear how this time dependency is captured (e.g., is this random, is this known/unknown).  This is important to clarify. Furthermore, I have doubts about the definition of the function as such.  In (1), it takes sets (the set of all Delta_j, and the set of all p_i) as arguments, while the definition says that its input is from R^n.  I don't actually understand what is meant.
> >    -> If the authors mean the collection of all outputs of these functions, then I believe this should be a tuple/vector, e.g., indicated with parentheses (...), and it needs to be ensured that |\mathcal{J}| + |\machcal{I}| = n if I'm not mistaken.
> > 2) **Unclear assumption on time dependence:** In Sec. 2.2, it is stated that no prior knowledge about the time dependence of C_t and g_{a,t} is available, but assumptions need to be made.  First, this is contradictory (assumptions represent prior knowledge).  Second, and more importantly, it didn't become clear to me what assumption on the TIME DEPENDENCE is actually made.
> > 3) **Generality of problem formulation:** Sec. 2.2, "For simplicity, we will proceed with a single safety constraint (A = 1) and one unknown variable."
> >    -> In my opinion, it is misleading to first state a much more general problem in the problem statement to then treat only a special case.  I suggest to state the concrete problem (the "special case") that is actually solved in the paper, and then DISCUSS how it can be generalized.  Otherwise, I don't find it very transparent what is done in the paper.
> > 4) **Section on related work:** The inclusion of a section on related work is appreciated.  However, it is not complete.  For example, not all relevant works on Safe BO (which are mentioned in the introduction), are discussed in the respective section for Safe BO.  IMHO, this is misleading.  If there is a section on related work, it should capture all relevant related work.
> > 5) **Related work on time-varying BO**: Sec. 3.3, TVBO: It is appreciated that the authors now discuss prior work on TV-BO.  However, also a few references for the type of methods that are described as "While many TVBO techniques operate under the assumption that the rate of change in the objective
> >    function is both known and constant" should be included.  More importantly, the authors argue that a key difference of their approach to prior works ("reset approach") is that they retain past data.  This needs more clarification.  In general, if there is indeed a CHANGE in the black-box function, and one does not make any assumption on that change, then retaining past data can be dangerous.  These data may then be arbitrarily bad and thus mislead the BO and thus endanger safety.  Retaining past data is not beneficial in general.  Given that this manuscript is about "safe" optimization, this is an important point IMO.
> > 6) **Your answer to W6**: But what does it mean in practice to have an oracle? In practice, I don't see how such information would be available.  So would it render (this part of) the algorithm impractical then?
> >
> >
> >
> > Finally, allow me a comment on the revision itself.
> >
> > The revised manuscript showed only SOME of the changed parts in blue color, but NOT ALL. This made the re-review very tedious. With all due respect, the authors should take more care when preparing the revision.
> >
> > Along the same lines, if you are to submit a revised version of the paper, please provide a proper letter with more detailed answers to the above and my previous comments.  For example, just saying that you fixed some problems, without detailing WHAT you did requires me to dig out the details from the manuscript itself, which is (again) very tedious if you don't properly mark all changes.

---

> > > ### Author Response · Authors · 2023-11-30
> > > **Response to Reviewer's Comments (1/3)**
> > >
> > > Dear Reviewer,
> > >
> > > Thank you for your thorough review and constructive feedback on our manuscript! We deeply appreciate the time and effort you invested in providing detailed comments. We have submitted a revised version incorporating your valuable feedback. In this revised version, we have addressed each of your concerns, ensuring that all changes are clearly marked in red for ease of identification and review. Below, we present our response to each of your comments, detailing the specific amendments we have implemented based on your insightful suggestions.
> > >
> > > ## Problem formulation
> > >
> > > - **Clarification of time dependency in $C_t$:** In the previous version of the manuscript, the time-dependent behaviour of the cost function $C_t$ was not explicitly defined. To address this, we have revised the first paragraph of the Problem Statement and Background section to explicitly state that the time-dependent behaviour of $C_t$ remains unspecified (unknown) across iterations. This clarification is intended to acknowledge the dynamic nature of the cost function in our problem setting, without assuming any specific form of time dependency.
> > > - **Representation of function inputs:** You correctly pointed out an inconsistency in how the function $C$ was defined and used. Initially, $C$ was described as a function from $\mathbb{R}^n \rightarrow \mathbb{R}$, but in the optimization problem, it appeared to take sets as inputs. To resolve this confusion, we have redefined $C$ as a function from $\mathbb{R}^m \rightarrow \mathbb{R}$, where $m = |\mathcal{J}| + |\mathcal{I}|$. This change ensures that the dimensionality of the input to $C$ aligns with the total number of outputs from the white-box and black-box functions.
> > > - **Introduction of the vector function $v(x)$:** To further clarify the nature of the inputs to $C$ and the safety constraints $g_{a,t}$, we introduced the vector function $v(x) : \mathbb{R}^n \rightarrow \mathbb{R}^m$. This function constructs a vector from the evaluations of the functions $\Delta_j$ and $p_i$ at the decision point $x$. The revised optimization problem now uses $v(x)$ as the input to $C_t$ and $g_{a,t}$, making it clear that these functions take a vector of real numbers as input, rather than sets.
> > >
> > > ## Unclear assumption on time dependence
> > >
> > > - **Clarification of knowledge on time-dependence:** In the previous version of the manuscript, we stated that we do not have any prior knowledge of how $C_t$ and $g_{a,t}$ change over time (in subsection 2.2.), which seemed contradictory when followed by stating the need for assumptions. This confusion has been caused by our poor phrasing. To resolve this, we have rephrased the misleading sentence in the first paragraph of subsection 2.2 to clarify that while we do not have specific knowledge of the time-dependent changes of $C_t$ and $g_{a,t}$, we still establish some general assumptions that apply uniformly across all iterations. This revision is intended to distinguish between the lack of specific, detailed knowledge of time-dependent changes and the necessity of having some general, overarching assumptions for the sake of model formulation and safety assurance.
> > >
> > > - **Emphasis on uniform application of assumptions:** In the first paragraph of subsection 2.2, we emphasized that the assumptions made are uniformly applied across all iterations to ensure safety. This clarification is crucial to understanding that our approach does not rely on specific time-dependent knowledge but rather on consistent assumptions that hold true for each iteration.

---

> > > ### Author Response · Authors · 2023-11-30
> > > **Response to Reviewer's Comments (2/3)**
> > >
> > > ## Generality of problem formulation
> > >
> > > To address potential concerns about the applicability of our approach to scenarios with multiple black-box functions and safety constraints, we added a discussion section in Appendix B where this generalizability is discussed in detail. This addition helps to assure the reader that while our initial discussion focuses on a simplified scenario, the broader applicability of our approach is considered and addressed. We can summarize our discussion about generalizability here with the following key points:
> > >
> > > - **Extension to multiple black-box functions:** Our framework is designed with the ability to accommodate multiple black-box functions $ (|\mathcal{I}| > 1)$ by modelling each function through a distinct GP. This approach is exemplified in our experiments, where we successfully apply our method to scenarios involving multiple black-box functions, each represented by its own GP. These examples demonstrate the adaptability of our framework to handle multiple unknown functions.
> > >
> > > - **Handling multiple safety constraints:** The framework is also extendable to scenarios with multiple safety constraints $(A > 1)$. The assumptions made in Section 2.2 are applicable to each safety function $g_{a,t}$. The primary challenge in a multi-constraint scenario is the complexity of the optimization problem. However, modern solvers are capable of handling such problems, provided they remain computationally feasible. We acknowledge that identifying specific conditions and interrelationships among multiple safety constraints that might lead to infeasibility is an area for future research.
> > >
> > > Overall, this discussion section emphasizes the flexibility of our methodology, making it suitable for a wide range of applications involving complex optimization scenarios with multiple unknowns and constraints.
> > >
> > > ## Section on related work
> > >
> > > We have revised this section to more comprehensively cover the relevant literature in Safe BO and TVBO. The key changes are as follows:
> > >
> > > - **Expansion of Safe BO literature:}** We have broadened the discussion on Safe BO in the related work section by including studies and advancements mentioned in the introduction section.
> > > - **Inclusion of specific references in TVBO:** In response to your request, we have added specific references to TVBO methods that assume a known and constant rate of change in the objective function.
> > > - **Clarification on Retaining Past Data:** We have clarified our approach's distinction from existing TVBO algorithms, particularly regarding the retention of past observation data. We also acknowledge the need for future research to address cases where data shifts might render past data obsolete, addressing your concern about the potential risks in scenarios with significant changes in the black-box function, as next: ``In this work, we operate on the belief that past data retains relevance and utility in learning the black-box function.The decision regarding which data to retain, discard, or remember weakly depends on the characteristics of the experiments under study. For example, in some real-world problems, it is necessary to consider data shifts that may render past data obsolete. We leave the detailed investigation of such specialized cases for future work.``

---

> > > ### Author Response · Authors · 2023-11-30
> > > **Response to Reviewer's Comments (3/3)**
> > >
> > > ## Our answer to W6
> > >
> > > We understand that our use of the term "oracle" might have led to some confusion, and we appreciate you bringing this to our attention. To clarify, our intention was to convey that we can conduct experiments and gather noisy observations from these experiments. We understand that the general definition of an oracle does not align with this context. In response to your feedback, we have made the following revisions to our manuscript:
> > >
> > > - **In the second paragraph of Problem Statement and Background subsection:** We have made clarifications in the text to emphasize that the black-box functions are unknown and we obtain noisy observations for their evaluations at the decision point $x$, given next:
> > > > Conversely, the black-box functions $p_i: \mathbb{R}^n \rightarrow \mathbb{R}$ for $i \in \mathcal{I}$ are unknown, and we obtain noisy observations for their evaluations at the decision point $x$, $y_i = p_i(x) + \epsilon_i$, within the environment where our experiments take place. Here, $\epsilon_i$ is assumed to be independent and identically distributed (i.i.d) $R$-sub-Gaussian noise with a fixed constant $R \geq 0 $.
> > > - **Assumption 2.3 revision:** We explicitly state that we assume noisy observations are obtained for each black-box function:
> > > > Noisy observations are obtained for each black-box function $p_i: \mathbb{R}^n \rightarrow \mathbb{R}$, $i \in \mathcal{I}$, represented as $y_i = p_i(x) + \epsilon$, where $\epsilon$ is characterized as independent and identically distributed (i.i.d.) $R$-sub-Gaussian noise with a fixed constant $R \geq 0$.
> > > - **Line 12 in Algorithm 1:** We have replaced the statement of
> > > > Receive $y_{i,t}$ from oracle $\mathcal{O}_i$ for chosen $x^*$
> > >
> > >     with a more common statement in literature:
> > >      > Obtain noisy observation $y_{i,t}$ for chosen $x^*$
> > >
> > > In conclusion, we hope that our responses and the revisions marked clearly in red color within the manuscript adequately address the concerns and suggestions you have raised. We have taken great care to ensure that the revised manuscript is easier to review and that our changes are transparent and straightforward to track. We believe that these revisions significantly enhance the manuscript's clarity and rigor. We are grateful for your guidance in this process and eagerly await your feedback on this revised submission. Thank you once again for your valuable input and for the opportunity to improve our work.
> > >
> > > Kind regards,
> > >
> > > Authors of Paper1180

---

### Review · Reviewer_8pZm · 2023-10-23

**Summary Of Contributions:**

The paper proposes a online Bayesian method for online decision-making under constraints where both the objective and constraints functions are partially unknown. The proposed algorithm uses Gaussian processes to estimate confidence bounds for the unknown parts, which can be used for exploration while ensuring that the constraints are satisfied.

**Audience:**

No

**Broader Impact Concerns:**

Not applicable here.

**Claims And Evidence:**

Yes

**Requested Changes:**

The overall formulation seems to be a bit vague to me (e.g., system characteristics, \Delta(x)). I think it would be helpful to give some examples or give more formal definitions.

It is not clear to me why Lemma 2.1 is needed, which seems to be quite straightforward.

Notations should be defined (e.g., Eq. 6 or L_2(D) in page 4).

References to the lemmas should be corrected. They are currently referred to as theorems.

There are quite a few typos and problems with English:
bottom of page 3: "we continue as such we have one"

**Strengths And Weaknesses:**

STRENGTHS

The main novelty of the paper is the formulation of a noisy optimization problem where both the objective function and the constraint functions depend monotonically on some unknown characteristics.

A heuristic method is proposed for safe exploration.

The proposed method is evaluated on two domains and compared to a few safe online optimization methods.


WEAKNESSES

With the monotony assumption, the proposed approach seems to be a straightforward extension of existing methods.

The proposed approach seems to be mostly heuristics. A theoretical analysis is missing.

The writing and the mathematical formalization could be improved (see below).

---

> ### Author Response · Authors · 2023-10-27
>
> Dear Reviewer 8pZm,
>
> Thank you for your time and effort in reviewing our work. We provide detailed clarification to address the issues raised in your comments below:
>
> - **Weakness 1 (W1):** We thank the reviewer for sharing their concerns regarding the novelty of the paper. We would like to clarify our distinct contributions, which can also be reviewed in the updated version of the manuscript at the end of the Introduction:
> > - We tackle a safe optimization problem wherein the objective and safety constraint functions are crafted incorporating both first-principle-based known functions and black-box functions. This formulation extends beyond the typical black-box optimization framework, transitioning towards a grey-box optimization paradigm which is notably relevant in industrial scenarios.
> > - A key part of our novel framework, Adaptive Real-Time Exploration and Optimization (ARTEO), is the simultaneous consideration of expanders-maximizers within the safe Bayesian Optimization (BO) setting, as opposed to the conventional separate iterations. By weaving uncertainty directly into the objective functions, we introduce a mechanism to fine-tune the trade-off between exploration and exploitation via a hyperparameter. Importantly, setting this hyperparameter to zero ceases exploration, thereby ensuring safety particularly when past explorative actions led to constraint violations. This strategy obviates the necessity for a distinct acquisition function to steer the exploration process.
> > - We have undertaken empirical validation of our methodology across both low-dimensional (with $2$ decision variables) and high-dimensional (with $200$ decision variables) safety-critical scenarios. The results affirm successful optimization devoid of safety infringements. Additionally, through empirical comparison of cumulative regret with established techniques like Safe-UCB and SafeOpt, we showcase that our methodology results in lower cumulative regret, underscoring its effectiveness as illustrated in Figure 1.
>
>     We believe these contributions exhibit a meaningful advancement in addressing the outlined safe optimization problem and hope that the revisions and clarifications in the manuscript elucidate the novelty and practical significance of our work.
>
> - **W2:** The safety guarantee of our algorithm is built on a theoretical analysis as explained in the Problem Statement and Background section. On the other hand, our work misses theoretical regret analysis. [As we respond to the reviewer AWtR] We agree that providing regret and sample complexity analysis would be an interesting addition to the paper. However, the proposed algorithm works with an extended objective function $f_t$ composed of the original cost $C_t(\cdot)$ and the uncertainty quantification $U_t(x)$, scaled by the hyperparameter $z$. Thus, the solution of the optimization problem with $f_t(\cdot)$ will depend on the numerical solver. The impact of the solver and its numerical properties on the solution is an addition planned for future work. For now, we have provided the empirical regret analysis for our first case study in the paper (Figure 1 and subsection 4.1.2).
>
> - **W3, Requested Change 1 (R1), RC3 \& RC5:** We have made a major revision to sections 2 and 3 to clarify and simplify our problem formulation and proposed algorithm. We fixed the mentioned sentences and typos, thank you for pointing them out. We hope the revision addresses the concerns regarding the notation, formulation and writing.
>
> - **RC2:** Lemma 2.1 (in original, Lemma 2.6 in revised version) is helpful to show $g$ is continuous even though it is a nested function of an unknown (continuous) function $p$ since it is formed by algebraic operation over two continuous functions. If $g$ would be formed by a different relationship (not algebraic), we couldn't state $g$ is continuous and later it has a monotonicity-based relationship with $p$.
>
> - **RC4:** Apologies for earlier mistakes on references to lemmas and definitions. It is fixed in the revised version.
>
> If you have any additional questions or comments, we would be happy to further discuss.
>
> Kind regards,
>
> Authors of Paper1180

---

### Author Response · Authors · 2023-10-27
**Updated version submitted**

We thank all reviewers for their thoughtful reviews.

We have just uploaded an updated version of the manuscript, taking into account the suggestions from all the reviewers. We have tackled the weaknesses and requested changes by reviewers through substantial revisions of sections 1, 2, and 3. along with addressing minor points in section 4. Although "Compare Revisions" should highlight our changes, we have also used blue text for the changes mentioned in our answers for your convenience. In summary, we have undertaken the following amendments:

- Enhanced the introduction by better elucidating the motivation behind our work, and drew connections between our work and the highlighted research domains of Bayesian Optimization (BO), safe BO, and time-varied BO.
- Refined the contributions section by articulating more precise claims regarding our work.
- Revised the problem statement and background sections, making the notation simpler and clarifying how our paper diverges from existing methodologies.
- Expanded the experiments section by explaining challenges and the dimensions of the problem to offer a clearer understanding of the case studies.
- Adjusted the appendix to align with the new formulation and notation.

We hope the revision addresses the concerns regarding the propositioning of the algorithm, the novelty of our work, notation, formulation and writing.

---

### Decision · Action_Editor_pC4S · 2023-12-05

**Recommendation:** Reject

**Comment:**

The initial version of the paper lacked clarity, but the revised versions improved significantly the explanations.  This is good.  However, the question of when and how safety is ensured remains.  Theorem 2.11 indicates with what probability some value sampled from a GP may fall in a confidence interval, but the link between this theorem and the ARTEO algorithm is missing.  One would expect to see a theorem that states that ARTEO will violate the underlying constraint at most x% with high probability when finding the optimum of the objective.  However, there is no such theorem.  In fact, the proposed algorithm is really a heuristic with a hyperparameter z that is updated in a way that is unclear despite the pseudocode provided in Algorithm 1. There is also a lack of clarity in the experiment with the electric motor.  Equation 11 describes the safety constraint, but this is really confusing since the constraint refers to the mean and variance of the GP.  There should be some true underlying constraint independent of any GP.  Then the GP should express a distribution about the unknown parts of the constraint instead of the constraint referring to the GP.  Due to the lack of clarity, heuristic nature of the algorithm and missing link between the theory and the algorithm, this work is not ready for publication.

**Audience:**

Yes, this work will interest anyone interested in safe AI

**Claims And Evidence:**

The paper proposes a new technique for Bayesian optimization subject to safety constraints.  Here the safety constraints are partially unknown.  The paper claims that the proposed technique ensures safe exploration, however the proposed technique is a heuristic that lacks theory to guarantee safety or quantify the probability with which safety is achieved.  Empirical results in two domains suggest that the approach can ensure safety, but it is not clear when the partially unknown constraints will indeed be satisfied.